

# Marine data assimilation in the UK: the past, the present and the vision for the future

Jozef Skákala[1,2], David Ford[3], Keith Haines[2,4], Amos Lawless[2,4], Matthew J. Martin[3], Philip Browne[5], Marcin Chrust[5], Stefano Ciavatta[6], Alison Fowler[2,4], Daniel Lea[3], Matthew Palmer[1], Andrea Rochner[3,7], Jennifer Waters[3], Hao Zuo[5], Mike Bell[3], Davi M. Carneiro[3], Yumeng Chen[2,4], Susan Kay[1,3], Dale Partridge[1,2], Martin Price[3], Richard Renshaw[3], Georgy Shapiro[8] and James While[3]

[1]Plymouth Marine Laboratory, Plymouth, United Kingdom,

[2]National Centre for Earth Observation, United Kingdom,

[3]Met Office, Exeter, United Kingdom,

[4]University of Reading, Reading, United Kingdom,

[5]European Centre for Medium-Range Weather Forecasts, Reading, United Kingdom,

[6]Mercator Ocean International, Toulouse, France,

[7]University of Exeter, Exeter, United Kingdom,

[8]University of Plymouth, Plymouth, United Kingdom.

*Correspondence to*: Jozef Skákala (jos@pml.ac.uk)

**Abstract.**

In the last two decades UK research institutes have led a wide range of developments in marine data assimilation (MDA), covering areas from the MDA applications in physics and biogeochemistry, to MDA theory. We review the progress over this period and formulate our MDA vision for both the short-term and the longer-term future. We focus on identifying the MDA stakeholder community and current/future areas of impact, as well as the current trends and the future opportunities. This includes rapid growth of machine learning (ML) / artificial intelligence (AI) and digital twin applications. We articulate the MDA needs for future types of observational data (whether planned missions, or hypothetical) and what should be the response of the MDA community to the increase in computational power and new computer architectures (e.g. exascale computing). Although the specifics depend on the MDA area, we advocate for balanced redistribution of the new computational capability among increased model resolution, model complexity, more sophisticated DA algorithms and uncertainty representation (e.g. ensembles). We also advocate for integrated approaches, such as strongly coupled DA



(ocean/atmosphere, physics/biogeochemistry, ocean/sea ice) and the use of ML/AI components (e.g. for multivariate
increment balancing, bias-correction, model emulation, observation re-gridding, or fusion).

**1 Introduction**

Marine Data Assimilation (MDA) is the process of combining observations and model information to produce an estimate of the state of the ocean. Such estimates can provide a view of the history of the ocean (reanalysis), or provide the best available initial conditions from which predictions can be made. MDA is therefore a pillar of a "predictable ocean", one
of the major challenges addressed by the United Nations (UN) Decade of Ocean Science for Sustainable Development (2021-2030) (https://oceandecade.org/). At the same time, ocean reanalyses are essential benchmarks for climate studies and are used to assess trends in the state of the ocean and derived services. Furthermore, since data assimilation is a tool at the interface of modelling and observation, it can provide essential information across the disciplinary boundaries, such as informing observational scientists on observing network design, or modellers on how to improve model configurations,
forcing and parameterisations.

The UK plays a leading role in the international MDA community, hosting, or partly hosting, two major operational forecasting centres: the Met Office and the European Centre for Medium-range Weather Forecasts (ECMWF). The UK also has a strong reputation in data assimilation (DA) theory, e.g. provided by the Data Assimilation Research Centre (DARC) of the University of Reading. The influence of the UK community extends internationally through organizations such as
OceanPredict (including being instrumental in setting up the OceanPredict Data Assimilation Task Team and contributing to other Task Teams), its strong participation in expert groups (e.g. Mercator Ocean International DA expert group with impact on the Copernicus Marine Service) and through a range of international collaborations, such as the Met Office Unified Model (UM) Partnership, and a wide range of EU Horizon and ESA projects. UK MDA is also a critical part of systems used to generate ocean products exploited for national and international marine policy and services, including the UN Sustainable
Development Goals, the EU Marine Strategy Framework Directive, Blue Growth, marine safety, and national security (e.g. underwater operations).

The role of this paper is to review the developments of MDA in the UK in the last two decades, its state-of-the-art, and provide a unifying vision for the near and longer-term future.  We give an overview of the current and potential MDA stakeholders, review different UK MDA areas, and formulate a vision for the future of each of those areas. The vision also
reflects upon new, or currently accelerating areas, such as machine learning (ML)/ artificial intelligence (AI) and digital twins of the ocean (DTO), that could be combined with MDA for a substantial mutual benefit. Finally, we discuss desired developments in the infrastructure providing the resources for MDA, such as ocean observations, and the development of software and hardware used for MDA.




## 2 The UK MDA community and its stakeholders/beneficiaries

The UK MDA community includes DA scientists, as well as ocean modelling and observational scientists providing inputs to MDA development, affiliated to the nine centres shown in Fig.1. These institutions interact closely through the

National Partnership for Ocean Prediction (NPOP) and its Data Assimilation Activity Group. Some of them also interact through the National Centre for Earth Observation (NCEO), with NCEO providing additional links to the broader environmental (atmospheric, terrestrial) community. The areas of expertise of each institution are listed in Table 1.

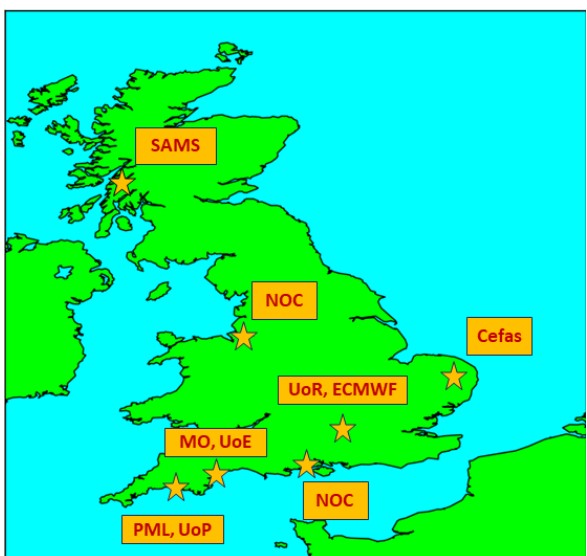

**Fig.1**: *The main institutions contributing to UK MDA and related efforts. The abbreviations are: University of Plymouth (UoP), Plymouth Marine Laboratory (PML), Met Office (MO), University of Exeter (UoE), University of Reading (UoR), European Centre for Medium-Range Weather Forecasts (ECMWF), National Oceanography Centre (NOC), Centre for Environment, Fisheries and Aquaculture (Cefas), Scottish Association for Marine Science (SAMS). The PML and University of Reading research is also done and funded as part of the NCEO. It should be emphasized that ECMWF is a European*

*institution rather than a UK one, but is partly based in the UK and has significant impact on UK MDA.*





| Institute | Physical model development | Physical DA | Biogeochemistry model development | Biogeochemistry DA | Coupled DA | DA theory | Observations |
|---|---|---|---|---|---|---|---|
| Cefas | | | ✓ | | | | ✓ |
| ECMWF | ✓ | ✓ | | | ✓ | ✓ | ✓ |
| Met Office | ✓ | ✓ | | ✓ | ✓ | ✓ | ✓ |
| NOC | ✓ | | ✓ | | | | ✓ |
| PML | | | ✓ | ✓ | ✓ | | ✓ |
| SAMS | | | | | | | ✓ |
| University of Exeter | | | | ✓ | | | ✓ |
| University of Plymouth | | ✓ | | | | ✓ | |
| University of Reading | ✓ | ✓ | | | ✓ | ✓ | ✓ |


**Table 1:** *The institutions involved in MDA in the UK and their areas of expertise in which they contribute to MDA. The Table marks the present situation as of 2024.*

UK MDA supports a wide range of stakeholder applications across the public and private sectors. It contributes to operational forecasts and reanalyses of key marine variables, both globally and regionally with higher fidelity, as well as to underlying scientific research. Key stakeholder applications of UK MDA are split below into end-user and scientific applications.

## 2.1 End-user applications

Real-time forecasts, initialised using MDA, are produced each day with various time ranges from a few hours to seasons ahead. Reanalyses are also produced which give information about the past state of the ocean. Ocean physics, sea ice, biogeochemistry, surface waves and weather information are all made available routinely to both specific users and the wider public. Existing and potential applications include:

- marine environment monitoring and prediction. This is of interest to national government departments (e.g. Department for Environment, Food & Rural Affairs) and agencies (e.g. UK Environmental Agency), local councils,



and industries including aquaculture and fisheries. Uses include the assessment of risk and planning the response to extreme events such as hypoxia, harmful algal blooms (HABs), and marine heatwaves. Products providing information about water quality and ocean health are also used, as are longer-term climate projections. Examples include spatial maps of oxygen deficient areas from Ciavatta et al (2016), which were included in an OSPAR assessment report on good environmental status (*https://oap.ospar.org/en/ospar-assessments/quality-status-*

*reports/*), and analysis of trends in marine heatwaves by Berthou et al (2024).

- marine safety and offshore industry (including energy and net zero) applications, such as beach safety, safe and efficient ship navigation, design and operation of offshore oil, gas and renewables, including providing ambient water characteristics for management of import/export capacity for UK energy sources using underwater cables and pipelines. Examples include work by Stephens et al (2018) and Copernicus products, such as *https://marine.-*

*copernicus.eu/services/use-cases/safe-transport-gas-north-sea*.

- coastal flooding forecasts to help prevent loss of life and infrastructure/property damage.

- near-real time products for national defence applications derived from variables including temperature, salinity, currents, and visibility.

- marine accident response including search and rescue applications, and marine pollution incident response.

Examples include pollution tracking system run by Cefas using ocean currents from regional analysis and forecast.

- climate change projections contributing to the IPCC reports. UK MDA already contributes to initialization of such projections, but also could in the future improve climate projections through better model parameter estimates, since a traceable set of models is jointly used for short-range forecasts, seasonal predictions and climate projections (e.g. Storkey et al., 2018).

- coupled ocean/atmosphere weather forecasts at short-range and seasonal timescales, and at global and regional scales. This includes forecasting events such as storms over the UK and Europe, tropical cyclones, monsoons, and El Niño events.

- range of very high-resolution coastal ocean operational systems, such as West Coast of Scotland Coastal Ocean Modelling System (WeStCOMS, https://www.sams.ac.uk/facilities/thredds/) and Western Channel Observatory

Operational Forecast (WCOOF) take boundary conditions from ocean analysis and forecast products. These can then feed into downstream systems such as HAB Reports (Davidson et al., 2021).

## 2.2 Scientific applications

As well as the users described in the previous section, there are both existing and potential scientific uses of products

generated through MDA, and the MDA systems themselves:



- use of reanalysis products to understand and monitor key climate metrics such as the variability and trends of the Atlantic Meridional Overturning Circulation (AMOC), sea ice extent and volume, and ocean heat, salt and carbon content. This includes reanalyses of ocean health indicators, such as pH to monitor ocean acidification, dissolved oxygen to identify trends in hypoxia, and net primary production to monitor biological productivity changes. Examples are decadal analyses of fluxes and indicators, including the phytoplankton community, in the UK regional seas (Ciavatta et al, 2016, 2018; Clark et al., 2020), as well as eco-regions and carbon fluxes in the Mediterranean Sea (Ciavatta et al, 2019).

- reanalyses are used to initialize future projections by ocean and climate modelling community (including the coastal modelling community) and as lateral boundary conditions to drive smaller-scale, regional models. Data generated by those projections then benefit the whole scientific community.

- realism and full data-coverage of reanalyses, as well as improved parameters and process estimates generated by MDA, support the community studying ocean processes (including scientific hypotheses testing) and metrics. Examples include improved understanding of North Atlantic circulation in Jackson et al. (2019). An interesting example inspiring future work can be a paper by Cole et al, (2012), who used reanalysis (however, produced by non-UK institutes) to identify impact of missing data on phenology metrics.

- reanalyses are also being used in the context of machine learning (ML) model development, where they have the advantage of providing gap-free structured training data (constrained by observations), instead of the intermittent observational products. Examples include emulators predicting marine oxygen (Skakala et al, 2023a) and ML model predicting marine nitrate (Banerjee and Skakala, 2024), both on the North-West European Shelf (NWES).

- improving model parameters using joint parameter-state estimation. This could feed into improved physical and biogeochemical (BGC) short-range, seasonal and climate projections, as well as underlying research applications. Examples include using 1D frameworks for parameter estimation such as the Marine Model Optimization Testbed (MarMOT, Hemmings et al, 2015) and the Ensemble and Assimilation Tool (EAT, Bruggeman et al, 2024), or estimating growth and mortality parameter variations in simple BGC models (Roy et al., 2012).

- reanalyses are also a great source of information on model performance and biases, which can feed into model development.

- MDA can also support sensitivity studies and help identify essential drivers behind specific processes. Examples include comparing the relative sensitivities of carbon flux estimates with respect to model configurations and assimilated variables at L4 station in the western English Channel (Torres et al, 2020).

- the products generated using MDA are underpinned by good observing systems. Making best use of the existing and past observing systems is one of the main motivations for the development of MDA methodology. This includes demonstrating impact of existing observations through Observing System Experiments (OSEs), influencing observational array design through Observing System Simulation Experiments (OSSEs), improvement



in satellite retrieval algorithms, especially in optically complex waters, navigating fully autonomous platforms into
regions of observational interest (reducing cost and carbon footprint), contributing to detection of problems with
observing systems in real time using automatic statistical quality control techniques, and investigating consistency
between different observational products. Since a whole section 5.4 is dedicated to this MDA application, including
a range of examples, we will leave the details for later.

## 3. The areas of UK MDA: their past, the state-of-the-art, and a vision for the future

### 3.1 Physical ocean and sea ice data assimilation

#### 3.1.1 Past and present


The UK has a long history of developing and running operational ocean forecasts with the first operational forecasts
of the Met Office's Forecasting Ocean Assimilation Model (FOAM) system produced in 1997 (Bell et al., 2000). This used
an assimilation scheme based on analysis correction (Lorenc et al., 1991; Martin et al., 2007). Simultaneously, an ocean
analysis system called System 1 (Alves et al., 2004) was developed at ECMWF, providing initial conditions only for the
seasonal forecasting system (Stockdale et al, 1998). It was developed around the Hamburg Ocean Primitive Equation
(HOPE, Wolff et al., 1997) model and employed an optimal interpolation (OI) scheme for assimilation of observations. The
ECMWF system grew over the subsequent years into a full 3D assimilation scheme, assimilating a range of data
(temperature, salinity, altimetry) with applications including also monthly forecasts (Balmaseda 2005, Balmaseda et al,
2008, 2009).

The Nucleus for European Modelling of the Ocean (NEMO) model was adopted at the Met Office around 2007
(Storkey et al., 2010) and implementation of the NEMOVAR data assimilation system (e.g. Mogensen et al, 2009) began at
the Met Office in 2011 (Waters et al., 2015).  The same systems were also adopted by ECMWF as part of their new ocean
reanalysis system (ORAS4, Balmaseda et al., 2013) that replaced HOPE. This was in 2016 further upgraded to the currently
used OCEAN5 reanalysis system, which is still based on NEMO and NEMOVAR (Zuo et al, 2015, 2017, 2019). The use of
these community systems for the ocean model and data assimilation has facilitated significant collaboration among UK and
European partners over this period.

Presently, in the UK physical ocean and sea ice data assimilation is primarily developed at the Met Office, ECMWF
and the University of Reading, with the underpinning NEMOVAR assimilation code being developed jointly by the Met
Office, ECMWF, CERFACS (Centre Européen de Recherche et de Formation Avancée en Calcul Scientifique) and INRIA
(Institut national de recherche en sciences et technologies du numérique). NEMOVAR ocean physics assimilation is



employed at the Met Office and ECMWF as a multivariate incremental 3DVar-FGAT (First Guess at Appropriate Time) scheme. It uses physical balance relationships to transfer information between physical ocean variables (Weaver et al., 2005) and employs an implicit diffusion operator to efficiently model the spatial background error correlations (Weaver et al., 2016). It includes bias correction schemes for sea surface temperature (SST, Balmaseda et al, 2007, 2013, While and Martin,

2019) and sea level anomaly (SLA) data (Lea et al., 2008). There were recently major new developments to NEMOVAR functionality through the implementation of capability to use hybrid ensemble/variational algorithms, including efficient methods for ensemble localisation. This functionality has been developed to increase flow-dependence within estimates of background error covariances to improve the quality of physics reanalyses and forecasts. It was applied in the Met Office global marine physics DA system where an ensemble forecasting capability was developed, and the impact of using the

ensemble information in the background error covariances in a hybrid-3DEnVar scheme was tested (Lea et al., 2022). Similar hybrid methods are becoming part of the upcoming ECMWF reanalysis system ORAS6 (see Fig.2). Finally, early versions of hybrid physics DA component have now been also developed for applications within the North-West European Shelf (NWES) forecasting system (Skakala et al, 2024).

Operational short-range forecasting systems are run at the Met Office and ECMWF for the global ocean and sea ice.

Furthermore, Met Office also runs regional short-range forecasts for the NWES, using high-resolution (1.5 km) coupled ocean-wave models. Global and regional ocean reanalyses have been also produced by the Met Office and ECMWF for many years, e.g. in support of seasonal forecasting. These include collaborative work with the University of Reading on altimeter assimilation, which is currently focused on improving the reanalysis products using smoothers (Dong et al., 2021, 2023).

Physical observations of the ocean and sea ice which are assimilated operationally include SST data from both in situ and satellite platforms, SLA data from satellite altimeters, sea-ice concentration (SIC) data from satellites, and in situ profiles of temperature and salinity from various platforms. Recent research to increase the variety of assimilated observations includes the development of assimilation of satellite sea surface salinity (SSS) data (Martin et al., 2019), satellite sea-ice thickness (SIT) data (Mignac et al., 2022; Fiedler et al., 2022) and preparation for the SWOT (Surface Water

and Ocean Topography) mission which provides wide-swath altimeter measurements (King et al., 2021).

Upcoming developments to NEMOVAR include a more efficient and up-to-date implementation of 4DVar capability (by INRIA). 4DVar is computationally expensive in large model configurations like the ones used operationally and has a significant maintenance overhead. However, it should bring significant improvements in performance, particularly in terms of making better use of observations, providing improved temporal consistency of outputs, and reducing shock in

the initialisation of forecasts, all of which are important aspects for most stakeholders. There is also work being undertaken in the NEMOVAR consortium aimed at developing multi-scale background error covariance matrices and to implement capability to represent spatial correlations in the observation errors (Guillet et al., 2019) which allow more information to be



extracted from high-resolution satellite data such as from SWOT. A studentship at the University of Reading (with funding from the Met Office) is currently investigating the control variables used to represent the horizontal velocities in ocean data

assimilation which should improve analyses of velocity, an important variable for many stakeholders.

### 3.1.2 Vision for the future

One of the main future goals is to harness the increase of computational capacity (e.g. exascale computing) to deliver (i) an efficient global ensemble hybrid-3DEnVar DA system at 1/12° spatial resolution (currently high-resolution

forecasts are initialised using lower resolution DA), and (ii) a shelf-seas ensemble hybrid-3DEnVar DA system at the 1.5 km resolution scale. The global high-resolution ensemble system would improve forecasts for a range of stakeholders, including the UK Royal Navy, marine navigation, as well as better coupled numerical weather prediction and seasonal forecasting.

Furthermore, it is essential to expand the observation types assimilated in physical MDA, e.g. this should include use of SWOT wide-swath altimeter data to improve initialisation of mesoscale structures, high-frequency (HF) radar,

Lagrangian drifter-derived velocities to improve velocity initialization and assimilation of sea ice thickness. There is particular emphasis on improving the sea ice analyses and the ocean velocities for a range of stakeholder applications, and the new types of observations should be combined with improvements in the ensembles and DA algorithms used in these two areas.

For regional models the long-term aim is to develop a separate 4DVar capability, which should allow us to improve

the quality of analyses of high frequency processes such as storm surges, tides and diurnal cycles, which are of interest to stakeholders near the coast. The 4D aspect will complement the use of ensembles in the estimation of background error covariances which capture the "errors of the day". This system has particular use for the Met Office, where wave and storm surge forecasts are generated without any DA. Both waves and surge models are highly influenced by the wind (waves, surge) and atmospheric pressure (surge). This means that the DA within these systems is unlikely to impact the longer

forecast range and suggests that the focus should be on improving the first 1-2 days of the forecasts (Saulter et al, 2020). Met Office thus plans to develop capability to assimilate data within these forecasting systems to improve such shorter-range forecasts, and to improve the representation of the ocean/atmosphere interface when waves become integrated in Met Office operational coupled forecasting systems.





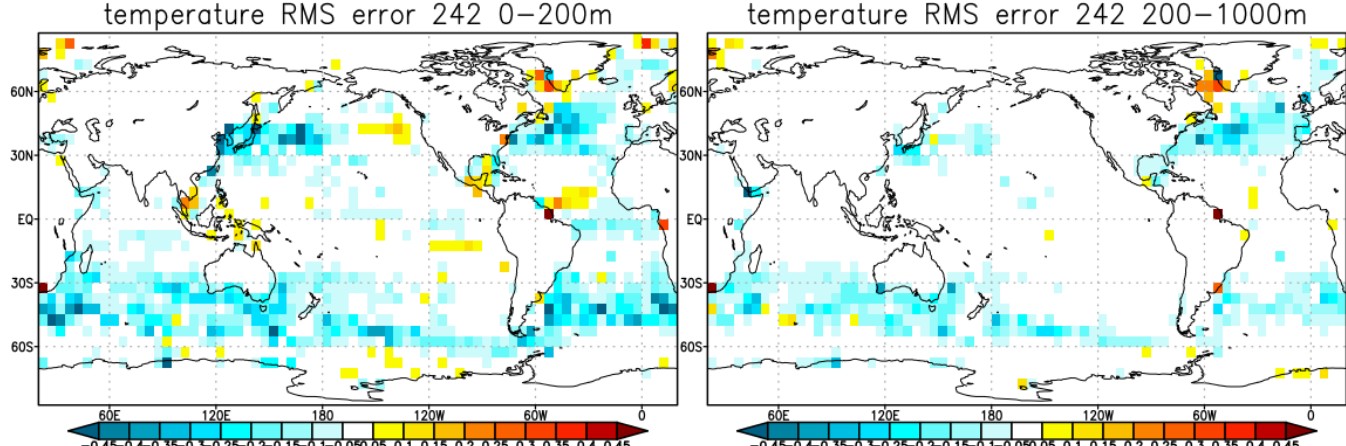

**Fig.2.** *Impact of using the ensemble of data assimilations to specify parameters of a hybrid background covariance matrix model (to be implemented in ECMWF system ORAS6) on the temperature first-guess forecast root mean square (RMS) errors in the top 200m (left) and in the 200-1000m range (right). The analysis is for the year 2017.*

The current reanalyses face a significant challenge due to the changing observing system over the reanalysis period, and due to responses of the model to the DA which sometimes introduces spurious signals that can contaminate the reanalysis products (e.g. spurious vertical velocities in the tropical regions). A range of model bias-correction techniques and data smoothing methodologies should be developed to improve the quality of ocean reanalyses, particularly in the period before Argo data are available, building on previous work by Zuo et al., (2019); Balmaseda et al. (2007); Bell et al., (2004) and Waters et al, (2017). This should allow more temporally consistent reanalyses to be produced, making them more suitable for climate and marine scientists studying past ocean changes.

Finally, we envision the physical ocean and sea ice DA to be transformed by machine learning (ML). The range of potential ML applications is ever increasing and varies from emulating components of the physical model or the tangent linear model to improve the system's efficiency, to development of hybrid models with a statistical model error correction component (Farchi et al., 2024). ML has also application in the data assimilation process itself, for instance, to provide an efficient emulator for NEMOVAR's diffusion-based correlation matrix model, or for diagnosing balance relationships between the analysed physical variables. Furthermore, wherever costly ensemble DA systems are used (e.g. at ECMWF), ML techniques could be developed to replace them with an emulator. Melinc and Zaplotnik (2023), developed 3D-Var using a variational auto-encoder (VAE) where the minimisation is performed in a reduced-order latent space discovered by VAE and the background-error covariance matrix is learned from historical data. Even more revolutionary scenarios can be envisaged. With the physical ocean being relatively slowly evolving and directly observed, and therefore being a simpler system than the atmospheric analysis system that deals with indirect satellite observations and evolves on much faster temporal scales, ocean physics analysis may be more easily replaced by an ML algorithm altogether. Recently developed





score-based data assimilation (Rozet and Louppe, 2023) and denoising diffusion model data assimilation (Huang et al.,
2024) show promising results.

## 3.2 Biogeochemical (BGC) data assimilation

### 3.2.1 Past and present

BGC DA research in the UK has mainly focused on state estimation rather than parameter estimation. Most past
research on the latter took place at NOC prior to about 2013 (e.g. Fasham and Evans, 1995; Hemmings et al., 2003, 2004),
with support from NCEO funding. This culminated in the development of the Marine Model Optimization Testbed
(MarMOT), a state-of-the-art tool for parameter optimisation in a multi-site 1D framework (Hemmings et al., 2015).
MarMOT is no longer being actively developed but is open source (https://projects.noc.ac.uk/marmot/) and available for
community use.

         For state estimation, two main strands of work have developed concurrently, starting to intertwine in recent years.
At PML, assimilation was developed for the complex European Regional Seas Ecosystem Model (ERSEM, Butenschon et
al, 2016), first in 1D (Allen et al, 2003, Torres et al., 2006), then for the western English Channel (Ciavatta et al., 2011) and
the whole NWES (Ciavatta et al., 2016). This used an implementation of the ensemble Kalman Filter (EnKF), with 100
ensemble members allowing multivariate updates, and 3D studies assimilating different products from satellite ocean colour.
Ciavatta et al. (2011, 2016) assimilated total chlorophyll, Ciavatta et al. (2014) assimilated diffuse attenuation coefficient
data, and Ciavatta et al. (2018) assimilated chlorophyll split into phytoplankton functional types (PFTs).

         Meanwhile, assimilation for the simpler Hadley Centre Ocean Carbon Cycle Model (HadOCC, Palmer and
Totterdell, 2001) was developed by the Met Office and NOC, applied to the global ocean. A sophisticated "nitrogen
balancing scheme" was developed to provide multivariate updates to non-observed variables in a computationally efficient
manner without ensembles (Hemmings et al., 2008). This was combined with an analysis correction scheme to allow
assimilation of chlorophyll from ocean colour (Ford et al., 2012; Ford and Barciela, 2017), optionally used with weakly-
coupled assimilation of physics data. In addition, a scheme was developed for assimilation of in situ $pCO_2$ data (While et al.,
2012). These schemes have since been applied with 3DVar using the NEMOVAR assimilation framework (Ford, 2020) and
in Ford (2021) they were applied with the Model for ecosystem dynamics, nutrient Utilisation, Sequestration and
Acidification (MEDUSA, Yool et al, 2013).  The latter study introduced assimilation of multivariate in situ profiles as might
be obtained from BGC-Argo data in an observing system simulation experiment using synthetic profiles. The work has been
extended in a collaboration between the Met Office and University of Exeter, assimilating biogeochemical observations from





various sources including satellites, ships, and BGC-Argo floats to investigate their individual and combined ability to
constrain the model's biogeochemistry (an example of this is shown in Fig.3).

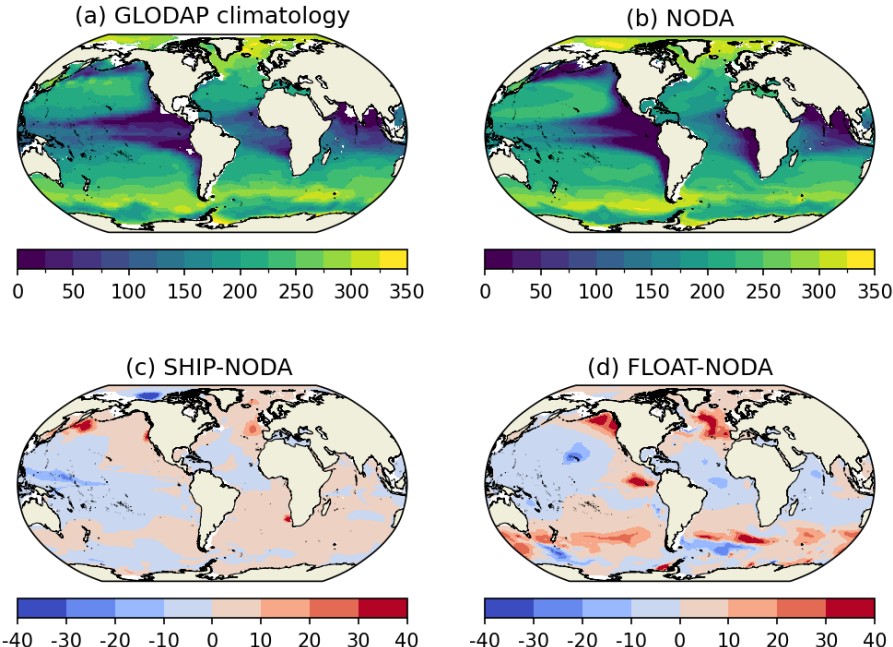

**Fig. 3:** *Impact of multi-platform BGC DA on oxygen (all in mmol/m3) at 200 m depth. (a) Oxygen concentration from the GLODAP climatology. (b) Oxygen concentration in December 2011 in a forced ocean-biogeochemical run, using NEMO-MEDUSA and forcing from ERA-Interim, without data assimilation (called "NODA"). NODA is initialised in 1980 from*
*climatology (EN4 for temperature and salinity, GLODAP for biogeochemistry). (c) and (d) show the difference in oxygen concentration in December 2011 to NODA when assimilating biogeochemical data in 2011 only. (c) Assimilation experiment called "SHIP", using in situ biogeochemical observations from GLODAPv2022 (oxygen, dissolved inorganic carbon, alkalinity, pH, nitrate, silicate, chlorophyll) and SOCATv2022 (fCO2). (d) Assimilation experiment called "FLOAT", using observations from BGC-Argo floats (oxygen, pH, nitrate, and chlorophyll).*

In parallel to this, a separate stream of work based on the Parallel Data Assimilation Framework (PDAF, Nerger and Hiller, 2013) is being developed for global NEMO-MEDUSA between PML and the University of Reading (funded by NCEO), mainly focusing on assimilation of carbon-from-space products. There is an effort to bring these parallel global carbon-oriented workflows closer together, exchanging important knowledge and capacity between the partners. Since 2016, Met Office and PML have collaborated on development of BGC assimilation for the NWES using NEMOVAR and the 7 km
resolution NEMO-FABM-ERSEM model. This builds on existing NWES physics and global BGC assimilation work at the Met Office and is informed by the previous NWES BGC assimilation experience of PML. The assimilation scheme is 3DVar



with a basic balancing scheme for phytoplankton variables, initially allowing assimilation of total and PFT chlorophyll from ocean colour (Skákala et al., 2018). Alongside physics data, total chlorophyll is currently assimilated into the Met Office operational forecasting system (McEwan et al, 2021), and PFT chlorophyll was assimilated within a multi-decadal

reanalysis product provided for Copernicus Marine Service (Kay et al, 2016). Further research activities have explored the assimilation of spectrally resolved PFT absorption coefficient data from ocean colour (Skákala et al., 2020) and combined assimilation of in situ chlorophyll and oxygen from gliders with ocean colour and physics data (Skákala et al., 2021). This new capability has been demonstrated in near-real time as part of a proof-of-concept of an autonomous and adaptive observing system (Ford et al, 2022). Tests have also been performed to assimilate BGC-Argo data in the same domain.

Furthermore, work to improve the background error variances and spatial covariances for ocean colour DA using the diagnostic tools of Desroziers et al. (2005) has been recently led by University of Reading (Fowler et al, 2022). Most recently, fully flow-dependent background error covariances in the form of a 3DEnVar system (following the global physics work of Lea et al, 2022) were implemented by Skákala et al (2024), with new versions of short-term reanalyses being tested and validated for Copernicus (Skakala et al., 2023b).

There are several ongoing developments for the regional NWES system. These include implementing the BGC assimilation in the high-resolution 1.5 km domain used for operational physics forecasting (Tonani et al., 2019), and refining the ERSEM representation of optics by including explicit representation of Colored Dissolved Organic Matter (CDOM), sediment, and their optical signatures. This latter work will also provide estimates of spectrally resolved reflectance, and assimilate hyperspectral reflectance data into the model, further strengthening the link between our modelling efforts and the

remote sensing algorithms of the EO community.

### 3.2.2 Vision for the future

The assimilation of BGC data poses specific challenges to MDA (Fennel et al, 2019): BGC models are much more complex and poorly constrained than the physical models, with BGC model parametrization being a particular consideration (e.g, ERSEM has over 400 mostly poorly constrained parameters). The complexity of biogeochemistry can also easily mean

that model parametrization hides a lot of internal dynamics (e.g. differences of internal species composition of a specific phytoplankton functional type represented by a single model state variable). Such differences can demonstrate themselves in spatio-temporal variability of recursive, or regionalized model parameter estimates, which are treated as constant in the models. Furthermore, BGC observations are typically harder to obtain and sparser than the physics observations, and may themselves have large uncertainties.

The current NEMOVAR-based BGC DA system is largely univariate, calculating increments for each observed variable independently, with limited number of non-observed variables subsequently updated via a balancing scheme, which for the operational NWES system is relatively basic. Assimilation is only routinely performed for chlorophyll from ocean colour, and the assimilation focuses solely on the state estimate. Our vision is to implement a fully multivariate data





assimilation scheme which integrates observations across multiple variables and observing platforms, including physics
information, whilst providing updates to model parameters alongside the model state. In NEMOVAR, fully multivariate
BGC assimilation can be either achieved through the use of hybrid ensemble-variational methods (building on the initial
univariate BGC tests of Skákala et al, 2024), or through incorporation into the NEMOVAR code of BGC balance
relationships, or through a combination of the two. It is likely that ensemble generation will need to be further improved by
both better tuning the current perturbations and including new sources of uncertainty, with some consideration given to
expanding the ensemble size whenever computational resources allow. More complex balance relationships could use as
their starting point the mass conservation scheme of Hemmings et al (2008), or focus on ML/statistical modelling, which is
being pursued in a current studentship between PML and the University of Reading.

Furthermore, new observational platforms like BGC-Argo and gliders, which can already be assimilated into our
models, are starting to deliver a much greater variety of BGC variables than we currently use operationally (e.g. carbonate
variables, nutrients, oxygen). It would be highly desirable to utilize such multivariate information and systematically
assimilate these datasets into our models. In situ BGC observations remain sparse though, and appropriate assimilation
methods should be explored to maximise the information gained from these datasets, including historical ones. Similarly,
multi-platform assimilation requires merging datasets across a wide range of spatial and temporal scales (including varying
depths), and there is a lot of room to rethink and improve the algorithms that presently do so. Further advances in
observation availability are expected through new hyperspectral satellite missions (e.g. PACE, Gorman et al, 2019), and
those data should be harnessed for assimilation to a maximal possible degree. This includes improving the optical
components of our models and bringing them closer to the water-leaving radiances seen by the satellite. Assimilation of
reflectance data can also complement the traditional chlorophyll assimilation in optically complex waters, where remote
sensing retrieval algorithms are less reliable. Improvements in optical components in our BGC models should also lead to
these models/MDA becoming regularly used to inform Earth Observation (EO) retrieval algorithm developments.

The state estimation in our BGC DA systems should be supplemented with estimation of BGC parameters, ideally
allowing for spatial and temporal variations in parameter values. Spatio-temporally varying model parameter estimation can
further improve model forecasts and contribute to better climate projections as well. To minimize spurious effects, such as
parameter estimates only compensating systematic model errors, joint state-parameter estimates should be combined with
model bias-correction (many BGC models have major seasonally varying biases, contradicting the assumptions made in DA
theory). One powerful approach to BGC model bias-correction might become ML (e.g. see Banerjee and Skakala (2024) for
discussion).

Finally, there is a clear need to implement DA in very high-resolution regional models, e.g. suitable for coastal
applications. This will ensure its suitability as an integral part of future DTOs (see Blair (2021) for overview). Physics MDA
has already been established for a 1.5 km resolution model on the NWES and similar efforts have already started for BGC





DA. A highly skilled, fully tested and efficient 1.5 km BGC DA system should be available in the near-future, and using this system for a wide range of higher resolution (e.g. DTO) applications will be desirable in the longer-term. The computational cost of such systems could be reduced through ML emulation, as outlined in more detail in Sec.3.5.2.

## 3.3 Coupled data assimilation

DA is often used in the context of coupled dynamics between different Earth System components, e.g. atmosphere and ocean physics, ocean physics and ice, or ocean physics and ocean biogeochemistry. The dynamical coupling between those components raises the question whether (i) separate DA solvers should be used for each component, with the assimilation increments from these separate DA systems being used to initialise a forecast of the coupled model, which is called ``weakly'' coupled DA, or (ii) the information about the coupling (e.g. cross-covariances) between the different components should be included into the DA system, which we call ``strongly'' coupled DA.

### 3.3.1 Coupled ocean physics-sea ice DA

Until now, both at ECMWF and the Met Office, ocean-physics and sea ice were analysed in separate minimisations as no cross-covariances were specified resulting to a separable computational problem. For its future systems, ECMWF is using a joint minimisation which will allow the possibility to specify background error covariance coupling terms, possibly through the balance operator. Simple post-processing balances will be applied in the next ECMWF systems such that sea ice increments will induce near surface ocean temperature increments, but not the other way around. Similar plans are being considered at the Met Office.

### 3.3.2 Coupled ocean-atmosphere DA

Weakly coupled ocean and atmosphere DA on global domain has been developed for many years at the Met Office (e.g. Lea et al., 2015) and similarly at ECMWF (de Rosnay et al., 2022). Since the early research and development (R&D) was carried out, an operational global coupled short-range forecasting system was implemented at the Met Office which delivered ocean products to the Copernicus Marine Service and demonstrated that such an approach was feasible in an operational setting (Guiavarc'h et al., 2019). This framework was then used as a basis for the implementation of a coupled ocean-atmosphere model and weakly coupled DA system for the main global weather forecasting system at the Met Office, which is now operational. Current work at Met Office aims to implement the ocean ensemble developments from Lea et al., (2022) into the coupled numerical weather prediction (NWP) ensemble (Lea et al., 2023). This allows improved uncertainty propagation from the ocean to the atmosphere through the forecast, leading to improved forecast uncertainties in both ocean and atmosphere. Further improvements are expected in the accuracy of ocean analyses from hybrid-3DEnVar DA in the ocean which would be enabled by these ensemble developments. Similarly to the Met Office, at ECMWF the operational



NWP has been weakly coupled since 2018, but advances towards stronger coupling have been made in coupled reanalyses, such as CERA and CERA-SAT (e.g. Schepers et al, 2018).

During a similar timeframe there has been significant research into coupled atmosphere-ocean DA at the University of Reading. This started with studying various aspects of strongly coupled DA algorithms in simplified coupled models (Smith et al., 2015, 2017, 2018, 2020, Fowler and Lawless, 2016). Recently there has been collaborations between the University of Reading and the Met Office under the WCSSP-India programme to understand the impact of coupled atmosphere-ocean DA on tropical cyclone prediction in the Bay of Bengal and how to improve the operational coupled DA

approach (Leung et al., 2022). There has also been work on understanding the nature of atmosphere-ocean error covariances from the Met Office coupled ensemble (Wright et al., 2024). Fig.4 shows an example result from that work, illustrating the variable nature of ocean-atmosphere cross-correlations, as estimated from the Met Office coupled ensemble.

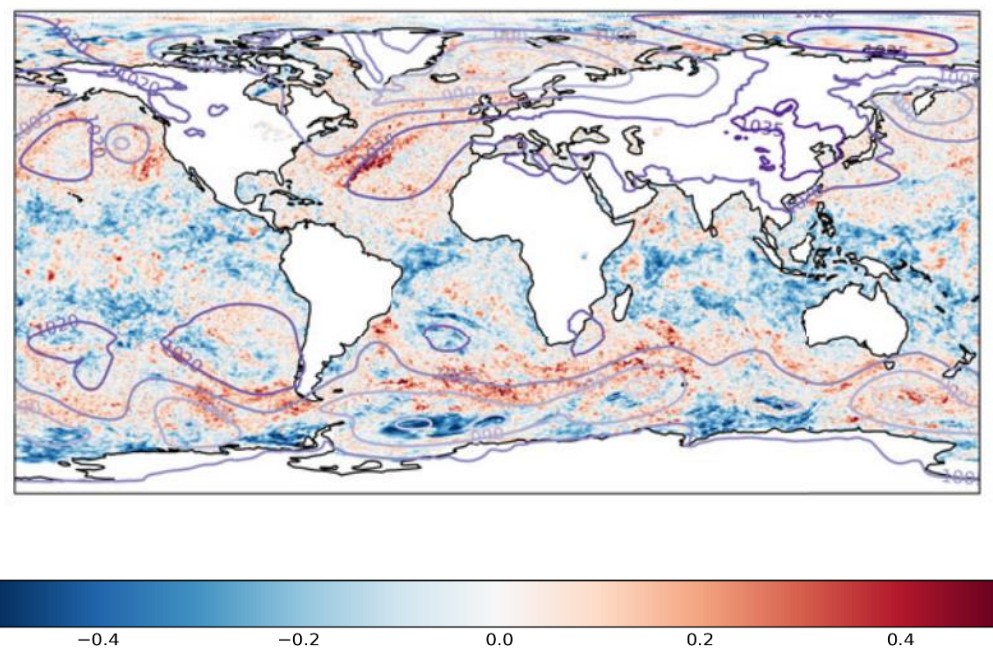

**Fig.4.** *Daily-mean correlations of sea-surface temperature and 10 m wind speed on December 5, 2019. Contour lines corresponding to the daily ensemble-mean sea-level atmospheric surface pressure field. Taken from Wright et al., (2024).*

A regional coupled ocean-atmosphere modelling framework has been developed as a UK collaborative programme for Regional Environmental Prediction (REP, Lewis et al., 2019). There are plans to move towards operational regional coupled predictions at the Met Office. This requires research into improved initialisation for the ocean component, including

ensemble initialisation, which is now beginning.





The main goal in the future is to establish a strongly coupled ocean-atmosphere DA to allow more information to be extracted from observations of both fluids. This is expected to improve the analysis in the near-surface ocean, as well as providing more consistent initial conditions for coupled forecasts at short-range and seasonal timescales. This can be achieved in a number of ways and through a number of different steps: (i) through continued exploration of the nature of

cross-fluid error covariances using ensembles and a decision on how these should best be included in coupled DA algorithms, (ii) the Met Office atmospheric data assimilation system is currently moving to use the Joint Effort for Data Assimilation Integration (JEDI) framework and Lea and Martin, (2023) demonstrated the feasibility of ocean DA using NEMOVAR code in the JEDI framework, which would allow more strongly coupled DA algorithms to be implemented, (iii) through exploration of the impact of the use of coupled observation operators to make use of satellite data sensitive to both

near-surface atmosphere and ocean variables, (iv) through improving methods to deal with the different timescales of the ocean and atmosphere in operational forecasting systems.

### 3.3.3 Coupled ocean physics-biogeochemistry DA

The standard physical-biogeochemical assimilation used by the Met Office within the operational model for the NWES, or on the global scale, is weakly coupled. It also typically uses one-way coupling between physics and BGC models

(no impact of simulated BGC state on physics). Currently, the inclusion of physics DA can degrade BGC model fields in equatorial regions (e.g. Raghukumar et al, 2015), and can have modestly detrimental impact on simulated phytoplankton in the NWES model (Skakala et al, 2022). Assimilating BGC data can compensate for this impact on the assimilated variables (Skakala et al, 2022), but the current systems do not fix the underlying issues, meaning non-observed variables can still be degraded, and biases can reappear during the forecast period. The combined impact of physics and BGC DA is also

dependent on the assimilation methodology (Nerger et al, 2023). Improvements have been relatively recently explored in both the global (e.g. Waters et al., 2017) and NWES system, where two-way physics-biogeochemistry coupling was included into the model (Skakala et al., 2022, see Fig.5). This was followed by some early attempts to introduce strong coupling within an offline balancing module used with NEMOVAR (Bertino et al, 2023), but these have proven to be unsuccessful. Therefore, the core issues with coupling marine physics DA with BGC remain unsolved, and are an open research problem

for the international MDA community (e.g. Raghukumar et al., 2015; Park et al., 2018; Gasparin et al., 2021).



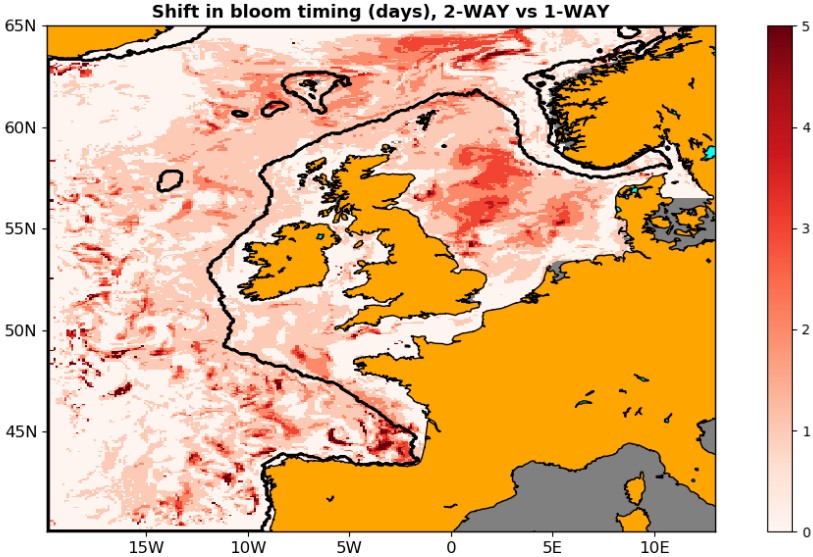

**Fig.5:** *Impact of two-way marine physics-biogeochemistry coupling on the timing of the phytoplankton spring bloom simulated by the NEMO-FABM-ERSEM model of the NWES. The one-way coupled model simulates late spring blooms, and*

*the two-way coupling partly corrects this by moving the bloom timing earlier towards the start of the year. The Figure is taken from Skakala et al (2022) and shows the number of days by which the bloom timing shifts when the two-way coupling is introduced into the model.*

A strongly coupled physics-biogeochemistry modelling and DA framework has the potential to improve simulations

by maximising the use of information and helping ensure physical-biogeochemical consistency (Anderson et al., 2000; Yu et al., 2018; Goodliff et al., 2019; Izett et al., 2023). This should be explored in future, with cross-covariances potentially derived using ensemble information in hybrid ensemble-variational configurations, or by using physical-biogeochemical balance relationships.

**3.4 Integrated observational science and systems**

Data assimilation can be used to aid observing system design and evaluation (Fujii et al., 2019), and as a component of "smart" observing systems (Ford et al., 2022), in the following ways:

(i) Current observing systems can be evaluated using observing system experiments (OSEs), sometimes referred to as data denial experiments (DDEs), in which different combinations of observation types are assimilated to assess the impact

on model analyses and forecasts of including or withholding certain observation types. With some assimilation methods, a





similar assessment can be performed using forecast sensitivity-based observation impact (FSOI), which assesses the impact each observation type has on improving forecasts in a single experiment (Eyre, 2021).

(ii) Proposed new observing systems, or changes to existing observing systems, can be evaluated using observing system simulation experiments (OSSEs), in which synthetic observations are generated from a model run representing a known truth, and then assimilated into a separate model configuration to assess the impact. This allows multiple potential scenarios to be compared.

(iii) "Smart" observing systems, in which ocean robots such as gliders autonomously adapt their sampling strategy based on ocean conditions, promise an efficient and cost-effective way of observing features of interest. Assimilative models are likely to play a key role in guiding these robots in real time, as part of a fully integrated observing and forecasting system. Such systems provide good examples of the emerging DTOs.

### 3.4.1 Past and present

Several OSEs have been performed using the Met Office's global ocean forecasting systems. Lea et al. (2014) ran a series of OSEs in near-real time, withholding different components of the global observing system in turn, and found a great deal of complementarity between the different observation types. King et al. (2020) withheld Argo observations in an OSE with the coupled ocean-atmosphere forecasting system and found significant degradation to ocean physics. The overall impact on atmospheric forecasts was small, but an impact was found on hurricane forecasts. Martin et al. (2020) performed OSEs adding recently available satellite sea surface salinity (SSS) data, which reduced salinity errors. This allowed a new set of requirements for SSS observation products to be developed. Mignac et al. (2022) compared assimilation of sea ice thickness from CryoSat-2 and Soil Moisture and Ocean Salinity (SMOS), and the impact on Arctic sea ice. As part of the ESA Climate Change Initiative, Ford and Barciela (2017) compared the assimilation of chlorophyll from different satellite ocean colour products, while Ford (2020) assessed the impact of assimilating different combinations of physics and biogeochemistry observations, and the consistency of information provided by each data type. A joint studentship between University of Exeter and the Met Office has assessed the impact of assimilating different BGC observation types (satellite ocean colour, BGC-Argo, ships) on reanalyses of the Southern Ocean air-sea $CO_2$ flux. These various OSEs have provided useful demonstrations of the benefits of the various observing systems to help ensure their continued funding. They have also improved our understanding of the ways in which the assimilation of particular observing systems performs, allowing improvements to the DA methodology associated with each observation type.

At present, as part of the ESA Climate Change Initiative, OSEs are being performed to assess the latest SSS products, and this will be extended to investigate the impact of SSS assimilation on carbon variables in a coupled biogeochemical model. The Met Office is also involved in the SynObs (Synergistic Observing Network for Ocean



Prediction) project as part of the UN Decade for Ocean Science to assess the impact of various observing systems as part of a coordinated set of OSEs. The project is led by the OceanPredict Observing System Evaluation Task Team (OSEval-TT). In the NWES BGC forecasting system, OSEs have been performed to assess the impact of physics assimilation on biogeochemical variables and assess the relative impacts of assimilating different ocean colour products. OSEs are also
being carried out in the NWES system to assess the impact of data from gliders, ships, and BGC-Argo.

Global and regional OSSEs have also been performed at the Met Office. As part of AtlantOS, Mao et al. (2019) participated in multi-system global OSSEs (Gasparin et al., 2019) assessing possible changes to Argo and moored buoy arrays, while Ford (2021) assessed the impact of assimilating different distributions of BGC-Argo floats. In a regional system, King and Martin (2021) ran OSSEs to assess the assimilation of wide-swath altimetry observations from the
upcoming Surface Water and Ocean Topography (SWOT) mission. This identified the need to effectively treat correlated errors. In unpublished work as part of the ABC Fluxes project, University of Exeter and Met Office collaborated to run an OSSE using simulated carbon and alkalinity data from the RAPID array in the North Atlantic.

In collaboration with Mercator Ocean international (MOi), the Met Office recently led an ESA project to carry out two sets of global OSSEs. The first set of experiments assessed potential future satellite observations of total surface current
velocities (TSCV), with results demonstrating a significant reduction in error for multiple physics variables (Waters et al., 2024a; Waters et al., 2024b). An example of the impact of the simulated TSCV data on the error in the drift of particles at the surface is given in Fig.6 which shows that assimilating TSCV data provides much improved estimates of the location of particles after drifting in the ocean. A second set of OSSEs compared the relative impacts of two wide swath altimeters versus 12 nadir radar altimeters (King et al., 2024). Finally, apart of OSSEs, similar, but more indirect methods can be also
used to inform future observational network design, such as observability metrics proposed for biogeochemistry indicators in Skakala et al, (2024).

MDA has also been used to aid ``smart'' observing systems/DTOs. Ford et al. (2022) designed and trialled an autonomous and adaptive observing system combining an ocean glider, an assimilative operational forecasting system, and a stochastic prediction model. This was deployed successfully as a proof-of-concept, tracking the spring bloom in the western
English Channel. Areas for further development were identified and are pursued in a present collaboration between PML, Met Office and the University of Exeter, tracking with multiple gliders harmful algal blooms and oxygen minima in a similar region in the western English Channel. The current mission is also based on a higher, 1.5 km resolution, model.



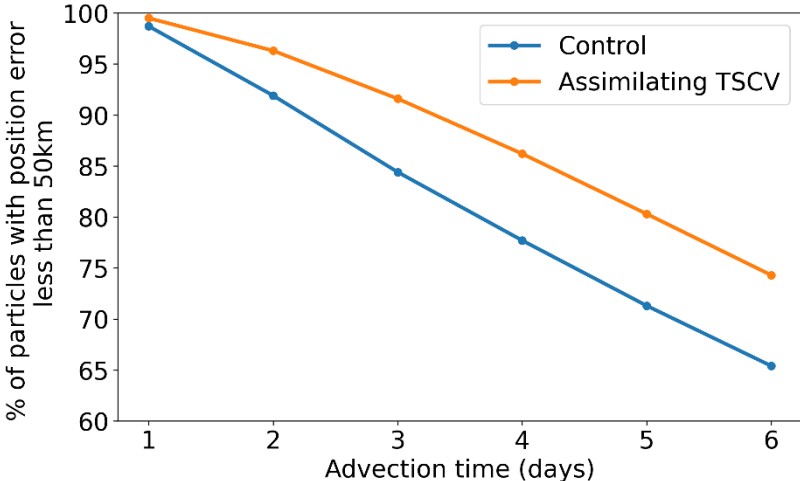

**Fig. 6.** *Impact of assimilating simulated TSCV satellite measurements on the global FOAM system. Particles were seeded every 1/4° and advected for 6 days from the 9th of September using the OceanParcels package with surface velocities provided by FOAM. The error in the location of the particles after each day was calculated and the percentage of particles with less than 50 km error on each day is plotted for two experiments: blue – the Control experiment with only the standard observing systems assimilated (SST, SLA, SIC, T/S profiles); orange – same as the Control but with TSCV data also assimilated.*

### 3.4.2 Vision for the future

The continuity of the existing observing systems requires advocacy from the MDA community to demonstrate their impact on operational forecasts and reanalysis. We therefore expect to continue to be involved in community projects like SynObs to show their impact to those funding the observing systems. We also expect to continue to use MDA systems to help design future observing systems through OSSEs.  The development of FSOI or a similar capability would allow us to provide more routine information about the impact of different observations in the assimilation, which should be very useful to both the observing and forecasting communities. More focus should be also turned to improving satellite retrieval algorithms. For example, assimilating reflectance data could help inform ocean colour algorithms in optically complex waters. Improvements can also be made using coupled ocean/atmosphere DA systems to retrieve observations of variables in both fluids from satellite measurements (de Rosnay et al., 2022)

A significant focus should be on ``smart'' autonomy as part of DTOs, through which MDA will likely further increase its contribution to optimizing the cost and efficiency of observations, bringing our science closer to net-zero goals. Such future DTO systems should also further optimize the way they integrate deterministic models with ML/AI and data-sources across a variety of platforms and scales.



### 3.5 Methods and theory relevant to MDA

### 3.5.1 Past and present

The development of DA methods and theory in the UK is driven by a wide variety of applications across the field of geophysics. Within the UK, DA theory research is mostly concentrated at DARC, based at the University of Reading, and is partly done in collaboration with the Met Office and ECMWF. Fundamental DA theory is also an underpinning theme to the research activities of NCEO, which seeks to advance the use of satellite data for understanding the carbon, water, and energy cycles. Overall, there is a strong partnership between DA theory and MDA applications in the UK, as demonstrated by a wide range of joint research activities, (e.g Fowler et al., 2022, Skakala et al., 2024, Leung et al., 2022; Wright et al., 2024, Dong et al., 2021).

Research in fundamental DA performed in the UK covers many different topics which have and can be transferred from different applications to MDA. These include developments in forecast and observation error covariance estimation (e.g. utilising consistency diagnostics, Fowler et al, 2022), non-linear DA algorithms (e.g. the development of parametric-free methods; Hu and van Leeuwen, 2021), coupling systems (e.g. improved estimation and treatment of in-domain and cross-domain covariances, Leung et al, 2022; Wright et al, 2024), bias-correction of model and observations (e.g. development of VarBC theory in Francis et al., (2023) and VarBC techniques in While and Martin, (2019)), reconditioning and preconditioning to improve convergence (e.g. Tabeart et al, 2020; Daužickaitė et al, 2021), development of simplified methods for smoothers (Dong et al. 2021, 2023), and metrics of observation impact (e.g. Fowler et al. 2020). An interesting theoretical development directly within MDA is a nested DA technique for high resolution modelling developed recently at the University of Plymouth (Shapiro et al., 2022, 2023). The UK community is also internationally leading on theoretical developments combining ML, modelling and DA, e.g. theoretical work led by Imperial College on neural assimilation (Arcucci et al, 2020), important theoretical research on combining ML with DA contributed to by University of Reading (Bocquet et al 2019, Brajard et al 2020a,b), work done at ECMWF on model bias-correction in the context of the 4DVar (Bonavita and Laloyaux, 2020, Farchi et al. 2021).

Some of the present DA research linked to MDA applications includes (i) a new technique for dealing with different timescales in coupled systems, developed at the University of Reading, and being tested at the Met Office as a method for treating late-arriving observations in ocean data assimilation, (ii) continued collaboration between the University of Reading and the Met Office to develop methods for treating the coupled ocean-atmosphere DA problem, including the incorporation of cross-covariance information within the assimilation process and the treatment of different timescales. It should be specifically noted that the transfer of the DA theory to MDA applications often relies on Early Career Ocean Professionals (ECOPs), happening through joint studentships, e.g. between the University of Reading and the Met Office, or PML.



Examples include a joint studentship involving the University of Reading and the Met Office developing a new control variable transform with a better treatment of balance in the velocity variables, or joint studentships involving University of Reading, PML and NCEO on implementing non-linear particle filter methods in BGC MDA, and developing an ML emulator for calculating corrections to non-observed variables as part of a DA BGC balancing scheme.

### 3.5.2 Vision for the future

A topic of particular interest for MDA theory is the ML-DA interaction. DA can be thought of as a physically constrained machine learning (ML) framework (Abarbanel et al, 2018, Bocquet et al, 2020, Geer, 2021). This, and the wide range of potential ML applications (for oceanography see reviews by Lary et al 2018, Sonnewald et al 2021) make it very attractive to be aligned with DA (for review of ML-DA methods see Cheng et al, 2023). Hence ML, which has seen an explosion of research activities due to the rapid progress in high-performance computing (HPC) and increase in data availability, will be an increasingly present theme in research developments of DA. Future ML-DA work should include developing ML emulators that dynamically downscale coarser resolution models into higher resolution, enabling us to emulate an ensemble of highly complex models (including applications in marine BGC) at very high spatial resolution at relatively low computational cost. A separate DA component should then assimilate high resolution data into the emulators, exploring non-linear DA methods. This type of downscaling would be in some way analogous to super-resolution work by Barthelemy et al, (2022), but different in that the ML in this case emulates directly the high-resolution model dynamics. As already mentioned in Sec.3.2.2, such high-resolution models are an ideal component to be implemented within DTO applications and, more broadly, many combined ML-DA theoretical developments will be needed to utilize MDA within the context of the DTOs. Other potential examples of ML applications within DA include model bias-correction, which has advanced significantly in recent years in numerical weather prediction (e.g. Farchi et al 2021), and use of ML to generate better observation operators able to account for poorly known processes.

There is a range of other theoretical developments that will be highly desirable in MDA, e.g. improvements in understanding of observation errors and new methods to account for observation error spatial and temporal correlations will allow for the full exploitation of the information available in observations (e.g. SWOT, King et al. 2021). Other examples include advancing smoothers to improve reanalyses by more effectively spreading the information from the observations, or development of better validation and tuning methods for ensemble perturbations.

### 4. Essential MDA infrastructure

The critical infrastructure for MDA includes the models, observations, MDA software, hardware and people. Although marine model developments are essential to improve operational forecasts, they rarely expand the MDA capacity compared to, for example, new observation types. The few cases, where we see potential for model advancement to improve



MDA, such as refining our bio-optical models, we have explicitly listed in Sec.3. Therefore, the following sub-sections will focus only on observations, MDA software, hardware and people.

## 4.1 Observations

Observations underpin the work done with MDA. Therefore, we provide in this section a detailed review of the types of observations available for MDA and formulate our future recommendations to the observational community.

### 4.1.1 The available observational types for MDA

The following observation types are currently available as resources for MDA:

- **Temperature.** Satellite observations of SST have been available for decades and their assimilation is well established in operational systems, including dealing with their biases. In situ SST measurements (e.g. from buoys, surface drifters and ships) provide good coverage globally and contribute to the strong impact of the satellite data as well as providing an unbiased reference for bias correction and validation (in the case of the drifters). Sub-surface measurements of temperature from Argo floats, gliders, moorings, ships (XBTs and CTDs), instrumented marine mammals and other platforms are routinely assimilated and provide reasonable global coverage to constrain temperature at large-scales. Improvements in measurements of deep ocean temperatures (below 2000 m depth), in western boundary currents and in marginal seas are being explored, the major area of weakness for the UK MDA community being the poor sampling of sub-surface temperatures on the NWES and adjacent ocean boundaries.

- **Salinity**. In situ measurements of salinity from Argo floats, moorings, gliders and ships have been available for many years, though the quality of salinity data can sometimes be less than that of temperature (due to drifts that can be difficult to detect in real time, fouling and inherent challenges in making accurate salinity measurements). Satellite sea surface salinity (SSS) data have been available since 2010, but their accuracy is generally low compared to in situ measurements, which makes it hard to gain benefit from assimilating them, particularly at mid- to high-latitudes. Issues with poor sampling of salinity in the NWES region are greater than for temperature due to extra difficulties in making accurate and long-term measurements.

- **Sea surface height**. Satellite altimeters have provided global coverage since 1993 but their number has varied. Recent sampling by altimeters has allowed reasonable initialisation of mesoscale structures in the deep ocean, but is not good enough to constrain some of the higher frequency processes of interest to many stakeholders on the NWES, e.g. surges, tides. The launch of SWOT is expected to improve the situation since it will resolve the SSH at high resolution within its swath, but the long repeat cycle of 21 days means that it is still not ideal for constraining all the scales we would wish.





- **Ocean currents.** Near-surface currents can be inferred from the positions of surface drifters which are usually drogued at about 15 m depth. These velocity measurements are not routinely assimilated into operational systems, though work is planned to make use of them. HF radars provide information on surface currents near some coasts, but there are few examples around the UK coast and no consolidated array. Proposals for future satellite missions to measure surface currents have been made, and these have been shown to have the potential to significantly improve the quality of ocean analyses and forecasts. Sub-surface measurements of currents inferred from Argo drifts are inaccurate and their assimilation has not been explored in the UK. Other measurements of sub-surface currents, e.g. from Acoustic Doppler Current Profilers (ADCPs), are sparse and often not sustained so their value for assimilation into operational systems is difficult to assess.

- **Sea ice variables.** Sea ice concentration has been observed by satellites for decades and is well established in operational assimilation systems. Sea ice freeboard and thickness data have been improving in their coverage and accuracy over recent years and their assimilation is well-developed in research mode (e.g. Fiedler et al., 2022; Mignac et al., 2022; Williams et al., 2023), but they are yet to be assimilated in operational systems. Very few in situ measurements are available of sea-ice thickness which makes the assessment and validation of model forecasts difficult.

- **Chlorophyll**. Satellite measurements of chlorophyll (total, or partitioned into PFTs) exist since 1997 and are available at high temporal frequencies and fine spatial scales. However, the data suffer from gaps due to cloudiness and low winter solar angle at high latitudes, and their accuracy and biases need to be better understood and accounted for in DA. However, satellite total chlorophyll is routinely assimilated into models and PFT chlorophyll assimilation is established as well. In situ measurements from e.g. buoys, gliders, BGC-Argo, ship measurements are still quite sparse but increasing in number all the time, particularly with the spin up of the BGC-Argo array and increasing use of gliders. Reconciling differences between in situ fluorescence and satellite ocean colour observations of chlorophyll remains a challenge.

- **Oxygen.** In situ measurements of oxygen have been made on BGC-Argo floats for a number of years and their number is increasing. Their assimilation has been developed and validated but is not currently operational. On NWES high-resolution oxygen can be provided by gliders, but quality control of oxygen data remains a challenge for operational applications (e.g. Skakala et al, 2021).

- **Nutrients.** In situ measurements of nutrients including nitrate, phosphate, silicate and ammonium are made by buoys, gliders, BGC-Argo and ships. While they are important variables in models which need to be constrained, their assimilation has so far not been a focus in the UK due to the sparseness of the data. However, nutrient DA is now starting to be developed, both globally as part of a studentship between University of Exeter and Met Office, and in NWES tests at the Met Office. Some extra benefit can be obtained from ML-derived nutrient products with good spatial and temporal coverage, such as the nitrate product from Banerjee and Skakala (2024), or the work of Sauzède et al. (2017). Finally, nutrient observations are very important for improving and validating models.



- **Carbonate variables.** There are rich in situ data available for pCO2 (or alternatively for CO2 fugacity, fCO2), which have been already assimilated (While et al, 2012) and shown to have a long-lasting impact on model fields. Some BGC-Argo floats measure pH, and direct measurements of dissolved inorganic carbon and total alkalinity are available from ships. A recent studentship between University of Exeter and Met Office explored the impact of assimilating these data on the Southern Ocean carbon cycle and found significant benefits, especially from BGC-Argo pH observations due to their widespread and year-round coverage. However, the potential of these data for assimilation remains largely unused and should be better exploited in the future. A further source of potential information is carbonate products derived from satellite observations of other variables such as salinity, temperature, and ocean colour (Land et al., 2015; Shutler et al., 2024), although it is unclear whether these will provide more benefit than assimilating the observed variables directly.

- **Other satellite-derived BGC variables**. Satellite-derived products for phytoplankton carbon, net primary production, coloured fraction of dissolved organic carbon, particulate organic carbon (both detritus and living), optical reflectance (hyperspectral), Kd (at specific wavelengths), spectrally-resolved light absorption by optically active tracers (e.g. PFTs), and even some regional products for zooplankton carbon and total dissolved organic carbon, exist (e.g. Brewin et al, 2021). In some cases, these products have been assimilated (light absorption, reflectance, Kd, e.g. Ciavatta et al, 2014, Skakala et al, 2020), and there are plans for assimilating others (phytoplankton carbon, hyperspectral reflectance). The quality and reliability of these products varies, but serious considerations (especially with upcoming model developments) should be taken for some of them to be used to complement the more standard chlorophyll assimilation. However, some issues with assimilating these products might need to be tackled, for example time-resolution of some of these products (e.g. satellite phytoplankton carbon) tends to be coarser than for the satellite chlorophyll.

- **Other in situ measured BGC variables:** Several other BGC variables can be measured by BGC-Argo, gliders, buoys, ships and other in situ platforms. This includes for example different optical measurements, carbon pools and fluxes, and phytoplankton and zooplankton biomass. Similarly to nutrients these observations are often too sparse to have been considered for assimilation to date, especially in near-real time, but have been used for validation. Potential for their assimilation exists though, especially for reanalysis. Furthermore, as in case of nutrients, advantage could be taken of products which use ML/AI to derive non-measured parameters from measured variables (e.g. Sauzède et al., 2017), and initial tests have been performed at the Met Office to assimilate these in the NWES system. Finally, a range of other biological data is, or is becoming, available, e.g. from omics, acoustics, plankton imagery. Currently it is much more reasonable to use these data for model development, calibration and validation, rather than trying to assimilate them into the present models, but their assimilation should be open for consideration in the future.





### 4.1.2 Planned observing missions and future recommendations

A list of planned satellite missions for Earth Observation is available from the World Meteorological Organization (WMO) (https://space.oscar.wmo.int/satellites). Of particular interest to the UK MDA community are missions which will improve observations of sea ice thickness (CIMR, CRISTAL and ROSE-L), begin measuring surface ocean currents (Harmony, ODYSEA), increase the sampling of small-scale SSH features through wide-swath altimetry, following on from SWOT (Sentinel-3 next generation, COMPIRA), and provide continuity of existing measurements through the Sentinel
programme.

       Improvements to the Argo network are also underway with the design of OneArgo focussing on three elements: improving the sampling by Argo in Polar sea ice zones and marginal seas; increasing resolution of Argo floats in the Western Boundary currents and equatorial regions; implementing more floats which measure biogeochemical variables (BGC-Argo) and which measure the deep ocean (Deep-Argo).

In terms of future need, the maintenance of the existing observing systems is paramount, but areas that require urgent improvement to allow UK MDA systems to better meet stakeholder needs include: (i) improving the sampling of sub-surface temperature and salinity in the NWES region, (ii) improving measurements of surface currents both globally and around the coasts of the UK, (iii) improving in situ measurements of sea ice thickness to complement satellite data, (iv) increasing the number of observed essential BGC and bio-optical variables, as well as increasing the number of sub-surface
observations, capturing biological features not seen from the satellites, such as deep chlorophyll maxima, (v) improving the accuracy and sampling of in situ measurements of the important BGC variables, alongside reliable uncertainty estimates, (vi) substantially increasing the number of coincident measurements of ocean and atmosphere, which could help assess the estimates of cross-fluid covariances for strongly coupled DA, (vii) use of ML to develop more complete products derived from observations. Addressing this long list of requirements, however, necessitates new investment and approaches. One
particular area we would like to highlight, particularly in the NWES environment, are autonomous ocean gliders. Gliders are increasingly being used to fill many of these requirements and have been demonstrated to work effectively in shallow shelf seas, filling a critical gap where Argo floats are largely ineffective. Gliders have the capability to provide sustained and regional scale (100s of km) measurements that cover almost the entire water column, from the surface to within a few metres of the seabed. Like many other marine science technologies however, gliders have mostly been deployed for short-term
process studies, despite international efforts to coordinate efforts to provide sustained capability (Testor et al, 2019). Gliders have however been shown to be cost effective for long-term multivariable monitoring of physical and biogeochemical states and change in UK seas (Loveday et al, 2022). Building on such demonstrators the Met Office invested in sustained operational deployments of ocean gliders in the North Sea from 2022, specifically targeted at improving sub-surface temperature and salinity on the NWES. The existence of such frameworks has the potential to provide a platform for further
expansion of broad-scale, long-term monitoring of the NWES, helping to link up the variety of ongoing monitoring efforts



from partner European states with autonomous mobile and adaptive measurement platforms. A number of research infrastructure initiatives have proposed frameworks around which to construct such coordination, including EU funded JERICO-RI and GROOMII programmes, but nothing yet exists to deliver funded, coordinated in situ monitoring of physical and biogeochemical states of the NWES.

## 4.2 MDA software

The main software systems currently being used by UK MDA scientists to carry out DA are listed below with a brief description of their uses:

- **NEMOVAR** (e.g. Mogensen et al., 2009) is a variational DA software developed as part of an international collaboration with significant contributions by the Met Office, ECMWF, CERFACS and INRIA. It supports both 4DVar and 3DVar versions, the latter using the FGAT method. NEMOVAR was originally developed for assimilating physical variables within the NEMO model, but more recently it has been adapted for BGC assimilation as well. It is used both on global and regional domains, with different BGC models (e.g. ERSEM, MEDUSA) and assimilating multi-platform observations. As already mentioned, NEMOVAR has recently been developed to allow the use of ensemble information, including hybrid ensemble-variational DA (Weaver et al., 2018), which has been applied for both physics and BGC. Several options have been added to NEMOVAR to reduce the computational cost including: i) the use of multi-grid algorithms (the horizontal error correlation modelling and localisation use the implicit diffusion operator which is a large percentage of the overall cost of the scheme and these can now be run on a lower resolution grid to reduce their cost), and ii) the use of mixed-precision (certain parts of the code can be run at reduced precision to lower the cost without significantly affecting the results). Work is also progressing to allow NEMOVAR to be run efficiently on future HPC architectures such as GPUs.

- **PDAF** (Nerger and Hiller, 2013), is an open-source framework for ensemble DA. PDAF simplifies the implementation of DA systems with existing models by providing tools to perform ensemble simulations on parallel computers. In addition, PDAF provides ensemble DA methods, in particular variants of ensemble Kalman filters and nonlinear filters like the nonlinear ensemble transform filter and particle filters. These methods are fully implemented, optimized, and parallelized so that they can make use of supercomputers. PDAF also provides tools for ensemble diagnostics and ensemble generation. PDAF has been coupled to both NEMO3.6 and NEMO4.0 in a collaboration including University of Reading, PML and NCEO. It has been also implemented in a public software EAT (Bruggeman et al, 2024) to be exploited in 1D applications. PDAF is currently used by PML-NCEO for 1D state and joint state-parameter estimation with GOTM-FABM-ERSEM and globally with NEMO-FABM-MEDUSA in NCEO marine carbon cycle programme. PDAF is also used by the University of Reading – NCEO with a coupled ocean-ice model (Williams et al, 2023).





- **JEDI** is an open-source framework for DA with development coordinated by JCSDA in the USA with numerous major institutions in the USA and elsewhere contributing, including the Met Office atmospheric DA group. In contrast to the other software described here (which are written in Fortran) it is written in C++ and has a modular, model-agnostic approach. Once a model has been interfaced to JEDI, various DA algorithms can be implemented. These are largely variational algorithms, but ensemble methods such as the LETKF are also available. As mentioned earlier, a scoping study recently demonstrated at the Met Office that ocean DA could be carried out using JEDI, interfacing to some of the modules of the NEMOVAR code (e.g. the diffusion operator for modelling background error correlations).

- **PML EnKF** is a FORTRAN package of the stochastic Ensemble Kalman filter developed by Evensen (2003) adapted to the assimilation of a range of ocean-colour products (chlorophyll, phytoplankton functional types, spectral diffuse attenuation coefficients) into coupled physical-biogeochemical models. It has been applied with POLCOMS-ERSEM in multiannual reanalysis of the biogeochemistry in the NWES and Mediterranean Sea. Features of this assimilation system and of its typical applications include: Gaussian perturbation of the assimilated data, the easily configurable number of analysed biogeochemical variables (up to 50), the use of 100 ensemble members, localisations of the analysis, analysis in log-space, inflation of the state variables (see Ciavatta et al., 2011, 2014, 2016, 2018, 2019 for details).

In future there should be a more unified and robust MDA software framework for UK research and applications, as this guarantees simpler pull through of methods to operational systems and makes better use of the limited human resources available. However, we recognise that there are some good reasons why fundamental research and operational applications might have different requirements from a software tool. The fundamental research needs ease of use and simplicity, while the operational applications need computational efficiency, robustness, and highly tuned configurations. This means full unification of DA software in the future is not expected. We will encourage the use of a single tool to allow for research pull through to operational systems and aim to move in this direction with the use of JEDI/NEMOVAR, but will keep the use of PDAF going for many important R&D applications (e.g. parameter estimation). We however remain open to revisit this decision in the future in the light of ongoing discussions and the accumulation of experience.

## 4.3 MDA hardware

UK MDA researchers have access to various high-performance computing (HPC) facilities including the NERC funded ARCHER2 facility and the Monsoon2 facility hosted at the Met Office. Other facilities are also available for use such as the PML hosted CETO and GPU MAGEO and clusters at universities. At the Met Office a new HPC is currently being installed which will provide a large increase in computing resources for research and operational uses, and ECMWF have





very large HPC resources which can also be accessed for specific research projects. A Joint Ocean Data Assimilation
Programme (JODAP) was implemented to make use of Monsoon2 HPC resources for various projects associated with the
NPOP DA activity group and associated PhD projects. This allows access to the Met Office's system for running research
experiments for global and shelf-seas model configurations with both physics and BGC models and DA included. It has been
used for numerous studies and has led to pull through of improvements in the DA applied to operational configurations.

Over the next 5 years or so there will be increased access to machines which make use of GPUs as well as the CPUs
currently utilised by our research and operational MDA systems. Significant effort has been put in by the Met Office and
others to allow the NEMO and NEMOVAR codes to be ported to GPUs and run efficiently on such machines. The approach
has been to use the PSyclone software developed by the Science and Technology Facilities Council (STFC) to parse the
Fortran codes of NEMO and NEMOVAR in order to add directives to the code which allow the most computationally
expensive regions to be run efficiently on GPUs. This separation of concerns means that the underlying Fortran code does
not need to be changed (though some efficiencies to the codes have been identified through this process) and the use of the
code on different types of processors (CPU or mixed CPU/GPU) is separated from the scientific development of the code to
a large extent.

For the future we recommend to continue developing further the capability of our MDA codes to run on GPU
machines as well as improving the efficiency of the codes on large numbers of CPUs. We need to make sure we harness the
increased computing power in the future by optimally distributing it into increasing resolution of the models, their
complexity, sophistication of DA algorithms and the quality of ensembles/uncertainty representation. Finally, we will
maintain and improve the ability to run operational configurations on research machines like Monsoon2 and its successors,
so that researchers and PhD students outside the Met Office can run experiments with these realistic systems and improve
the pull-through of developments into the operational systems.


**4.4 People**

The number of scientists working in MDA research and development is somewhat limited, and we need to keep
bringing new scientists into the field as well as improving the training available to MDA scientists. There is a good
availability of a variety of training activities across the UK MDA community. Two partners from NPOP MDA (ECMWF
and DARC from the University of Reading) run a coordinated set of annual introductory DA training courses (occasionally
also offered under NCEO). There are also other courses on related topics (e.g. ensemble forecasting methods, satellite data
assimilation) offered each year by ECMWF. Training is occasionally offered as part of project dissemination, e.g. in the
context of a recent EU Horizon project SEAMLESS, which has developed a user-friendly DA software EAT for 1D ``toy''
models (Bruggeman et al, 2024) providing the opportunity for non-experts to develop practical DA skills. The University of



Reading have recently developed a free Massively Open Online Course (MOOC) to introduce scientists to the basic ideas of DA and reanalysis (discoverda.org)

Within the UK MDA community there is also a significant amount of PhD student supervision and often these are jointly supervised by different partners including PML, Met Office and the University of Reading. These activities help to bring in new scientists to the MDA field, as well as improving the collaboration between the different UK partners.

Internationally there are occasional efforts to organise summer schools, for instance as part of the OceanPredict community, which focus on training early career scientists and PhD students in the science of ocean forecasting (including MDA). However, the last one of these was in 2017 so, whilst the training material is available and still somewhat relevant and useful, an update to this would be beneficial.

We need to develop and maintain people's skills through (i) joint student supervision, including improving the
framework within which studentships are proposed, (ii) establishing opportunity for talented and motivated students to continue their career within the UK MDA community after completing their PhD, (iii) developing simple toy models and software tools (including their documentation) to enable the wider community to gain ``hands-on'' experience with MDA and (iv) upscale the existing MDA community in ML techniques and strengthen links with environmental ML centres of excellence.


## 5. Summary

Marine data assimilation (MDA) is a critical part of the process of producing forecasts and reanalyses of physical and biogeochemical ocean variables at global and regional scale. Numerous stakeholders rely on the information produced by UK MDA systems including those using operational forecast and reanalysis products for downstream applications
(including by UK government, defence and energy sectors), scientists who use the MDA systems to improve understanding of the ocean and climate system, and scientists who use evidence from MDA systems to decide on the design and funding of ocean observing systems.

Looking into the future, UK MDA will have a chance to benefit from many new opportunities, such as exascale computing, further acceleration of ML/AI, and new observing missions, which will also bring many challenges. Both the
opportunities and challenges to some degree depend on the specific MDA area and our future vision can be summarized as follows:

- Physical DA developments will enable high-resolution ensemble forecasts at 1/12° resolution globally and 1.5 km resolution regionally, making use of hybrid ensemble/variational DA methods. Improvements to sea ice DA will be made to address stakeholder needs in the Arctic. New observation types will be assimilated to improve





representation of mesoscale structures and surface current forecasts. 4DVar methods will be investigated to improve the representation of high-frequency processes (e.g. surges), particularly in regional systems. Model bias correction methods will be further developed to improve reanalyses. ML will be considered for a wide range of purposes, from statistical bias correction, emulating NEMOVAR diffusion-based correlation matrix, learning balance relationships to potentially emulating the full DA system.

•   Biogeochemical (BGC) MDA developments will improve reanalysis and forecasts by assimilating a greater variety of data using multivariate DA techniques and, through advances in optical modelling, MDA will become much more useful for the EO community. ML will play far greater role also in BGC MDA, e.g. correcting model seasonally varying biases, or helping with multivariate balancing relationships. DA will be used to estimate BGC model parameters, as well as their spatio-temporal variations, supplying this information to the climate projections.

For the North-West European Shelf, we will refine the BGC DA system spatial resolution to 1.5km, enabling the system to play an active role in the digital twins of the ocean (DTOs) of the ocean. BGC MDA will also benefit from many of the developments made for physics DA.

   •   Coupled DA will develop stronger coupling between ocean and atmosphere components to allow more information to be extracted from observations of both fluids and provide more consistent initial conditions for coupled forecasts at short-range and seasonal timescales. Stronger coupling will be also established between marine physics and biogeochemistry in the DA, and between ocean and sea ice.

   •   Observation deployment and design will be better integrated with MDA systems using fully autonomous "smart" observing systems. The capability to routinely demonstrate the impact of observations on forecasts will be developed. Observing System Simulation Experiments will be used routinely in the design of the ocean observing system.

   •   Theoretical developments in DA will inform much of the above work. Through theory development, ML will become an essential part of the MDA process, including making (through emulators) the use of very high-spatial resolution ensemble forecasting with non-linear DA methods applied to highly complex models, computationally affordable. Overall, integration of ML with DA will make our MDA systems much more applicable in future DTOs. Smoothers will be further developed to improve the quality and temporal consistency of reanalyses. Methods to better estimate and integrate information about spatio-temporal observational error correlations will be developed, making the observations more useful for MDA.

The resources available for development of MDA in the UK will be developed in the following ways:

   •   Maintenance of the existing observing systems is paramount, but priorities for improved observations include better sampling of sub-surface temperature and salinity in the shelf-seas around the UK, improved measurements of surface currents both globally and around the coasts of the UK, improved measurements of sea-ice variables in the



Arctic, increased number of observed biogeochemistry (BGC) variables, as well as improved accuracy and sampling by in situ BGC measurements.

- The software used by the UK MDA community should be consolidated into two systems (JEDI/NEMOVAR and PDAF), with the potential of reducing this down to one system if useful. Work will be carried out to make these systems run efficiently on GPU-based computers. Large HPC resources will be made available which are shared between academic and operational partners to allow continued joint work on realistic configurations and improve pull-through of developments into operational systems.

- New scientists should be brought into the MDA community through joint studentships between the MDA partners and we should maximise the opportunity for them to continue their career within the UK MDA community. Improved software should be made available to train students and other researchers. The skills related to ML should be improved within the MDA community.

**Code/data availability**: No new data, or code published in this paper.

**Author contributions:** JS led the writing of the manuscript and all the authors contributed ideas, text and Figures.

**Competing interests**: The authors declare that they have no conflict of interest.

**Acknowledgments:** This work was financially supported by the UK National Partnership for Ocean Prediction (NPOP) and developed through the NPOP MDA activity group.

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
