# Peer review of "Marine data assimilation in the UK: the past, the present and the vision for the future"

_EGUsphere, 2024_

## Author Comment (AC1)

We would like to sincerely thank the reviewer for their positive comments, which were of great value. Please find our responses below (reviewer comment in bold font, our response in standard font and the suggested changes in italic font):

**1. L 110; L115; L135; L150, L630-660; L670  : references could be added to ensure consistency with the rest of the article.**

We thank the reviewer, the following references/links have been added to address the issues:

**L110:** here we don't have a specific reference, but the following weblink can be provided: "*https://www.cefas.co.uk/science/emergency-response/*"

**L115:** we will provide the following reference:

*Guiavarc'h, C., Roberts-Jones, J., Harris, C., Lea, D.J., Ryan, A. & Ascione, I. (2019) Assessment of ocean analysis and forecast from an atmosphere-ocean coupled data assimilation operational system. Ocean Science, 15, 1307–1326. Available from: https://doi.org/10.5194/os-15-1307-2019*

**L135:** we will provide the following references:

[revised manuscript text omitted]

**2. L340, L880 : EO has not been introduced, and it seems a little odd to mention the Earth Observation community only there in the summary. It could also be explicitly mentioned in the paragraphe on observing systems design L893**

We thank the reviewer for spotting this and we will introduce EO already on the line 340 and also add explicit mention of EO on the line 893, as requested.

**3. What we see on Fig 2 is not really explained.**

**What is error 242? (see title of the figure), does blue mean an error reduction?**

The "242" in the Figure title is a typo and will be removed. We will change the Figure caption to better explain the content of the Figure, please find the suggested text below:

*"Change in temperature Root Mean Square Errors (RMSE) between two experiments. The reference experiment uses parameterized background covariance matrix model, the second experiment uses the ensemble-based hybrid background covariance matrix model. Temperature RMSE is computed using model short-range forecasts against all in-situ observations in the upper 200 m (left) and in the 200-1000 m range (right), for year 2017. Negative values show improvement when using ensemble-based hybrid background covariance model."*

**4. L698: Kd has not been introduced**

Thank you, we will introduce it.

**5. L244 : It may be easier to follow by adding one sentence, maybe on the time scales of the impact of the atmospheric forcing.**

Thank you, we will address this by the text below:

*"Both waves and surge models are highly influenced by the wind (waves, surge) and atmospheric pressure (surge). Saulter et al. (2020) showed that assimilation of data into a regional wave model using NEMOVAR improved the forecasts over lead times of up to 12 hours, but errors in the surface forcing and wave model parameterisations dominated the forecast errors beyond 1-2 day lead time. The Met Office thus plans to develop capability to...."*

**6. L426-427 : a couple of sentences could explain explain what we see on Fig4.**

Thank you, we propose to add the following text to the Fig.4 caption (we think it is very specific information to be discussed in the main part of the text):

*"In the tropics we see negative correlations associated with warm SST and low wind speeds, which are linked to diurnal variations in solar radiation, with correlations strengthening as the ocean surface warms throughout the day. In contrast, the significant positive correlations of SST with 10 m wind speed in the North Atlantic are linked with strong SST gradients and tend to be associated with areas of stronger winds, the location of which varies synoptically. These areas of larger ocean/atmosphere correlations in mid-latitudes were shown to extend vertically into the ocean, throughout the mixed layer (Wright et al., 2024)."*

**7. L455: "the combined impact of physics and BGC DA is also dependent on the assimilation methodology": maybe one or two examples would help to link with future tests and potential improvements.**

We propose to expand the sentence as follows:

*"The combined impact of physics and BGC DA is also likely to be dependent on the assimilation methodology (Nerger et al, 2023), for instance how alignment of fronts and other features is considered (Anderson et al., 2000; Yu et al., 2018), and how increments of different variables are projected onto different scales (Waters et al., 2017)."*

**8. L 473: as there are slightly more details about this in 3.2.2 the authors may refer to this section**

Thank you, we will do so.

**9. L1040 : inconsistency between Eyre et al (2021) in the text, and (2022) in the references**

Thank you, we will correct this.

**10. L 1188 : add the year in reference Nerger et al (2023)**

Thank you, we will address this.

Best wishes,

Jozef Skakala and the co-authors

---

## Author Comment (AC2)

We would like to thank the second reviewer for their considered comments even if their overall evaluation of our manuscript was negative. To respond, we would like to firstly highlight what seem to be the main areas of the reviewer's concern, and then respond to these points one by one.

The main concerns of the reviewer seem to be focused around the following points:

1. **The lack of clear reason why the UK MDA community should write their own review and vision paper, rather than just contributing to the standard international community papers (i.e. driven by OceanPredict).**

We did not expect the paper's topic to be challenged at the review stage, as the submissions (with their exact topic) to Ocean Science's (OS) special issue "OS Jubilee: reviews and perspective" have to be first consulted with the OS editors, approved and subsequently invited by the editorial board, which was also our case. Nevertheless; not just national, but even Institution-specific papers, describing research plans and vision, are commonly published in OS, e.g. see a Met Office focused paper as an example (Siddorn et al, 2016, https://-os.copernicus.org/articles/12/217/2016/). This suggests that OS finds such type of papers to be of interest and we expected this to be automatically considered by the reviewer.

However, let us emphasize the reasons why we decided to write this paper:

a) The UK Marine Data Assimilation (MDA) community is not just a collection of different institutes, but a very close collaborating entity. Indeed, the paper is full of examples when at least two of the UK partners contributed jointly to a certain development, and often the development was by even more partners than two. We are sorry if this aspect has not come across clearly enough, and we are very happy to emphasize it better in the revised version of the manuscript. It is worth mentioning that the paper was written by an entity facilitating this close collaboration, by the UK National Partnership for Ocean Prediction (NPOP) MDA group.

b) A vision for MDA needs to be shared by a team with the resources to implement it. We have tried and failed to agree a vision at a European level within Mercator-Ocean International (MOi). NPOP was set up to develop shared ocean prediction capabilities for the UK so it is entirely appropriate for NPOP to have its own vision for shared MDA systems. It is highly desirable for such a vision to be peer reviewed and published. It takes a substantial effort to develop such a vision and an articulation of the arguments supporting the vision that has been developed is likely to be of interest and value to groups in several countries.

c) We also believe that focus on a closely collaborating community representing a country that contributes importantly to international MDA developments is of wide interest, as it enables one to go into far greater detail than would be ever possible in an international review covering the global community. Even though such detail reflects a specific country, we believe it is interesting from the international point of view as well, as many aspects of it might have analogues in other countries. Furthermore, the paper also demonstrates the substantial level to which the UK work feeds into international developments (e.g. on NEMOVAR, or within the many international collaborations with partners such as CERFACS, AWI, MOi, NERSC and many others).

d) The ability to focus on far greater details is one of the reasons why we have gone in this paper much further than what has been summarized in the existing international community review papers mentioned by the reviewer (e.g. the ones from 2019 issue of Ocean Predict, such as Moore et al, 2019, https://doi.org/10.3389/fmars.2019.00090, Fennel et al, 2019, https://doi.org/10.3389/fmars.2019.00089). Examples include the detailed mapping of stakeholder use of UK MDA products, a detailed section on the observational requirements of the UK MDA community and detailed discussion of our hardware and software needs. Many of these sections are also meant to facilitate discussion across different scientific communities, from modelling to ocean observation scientists, with the expectation that they will hopefully be read by many scientists outside of MDA. Furthermore, unlike the OceanPredict reviews, our approach enables us to discuss the different MDA areas alongside each other, which provides an interesting opportunity to compare and contrast the differing needs and requirements across the different MDA topics, as well as find similarities.  Such comparison is inherently contained within the bullet points presented in the Summary section, but we are happy to make it more explicit during the paper revision.

e) Finally, the paper opens multiple new and timely topics not discussed in the previous MDA review papers, such as machine learning applications in MDA and the digital twin systems. This extra focus was possible thanks to the UK community being the source of many pioneering developments in those fields.

f) Overall, we believe this paper to be complementary to the international reviews the reviewer mentions, with both types of reviews worthy being published.

**2. That there is not enough close collaboration between the UK Institutes to justify this paper.**

There are many examples of our close collaboration across the manuscript, e.g. large number of developments in both physics and air-sea coupled DA have been made jointly between the Met Office and the University of Reading, most of the developments in marine biogeochemistry DA have been done jointly by PML and the Met Office, with additional important contributions from the University of Exeter and University of Reading (with University of Reading providing the theoretical underpinning for wide range of applications developed at the other partners). This is documented by the papers: e.g. the manuscript references provide for the period of the last 5 years around 20 joint publications including multiple UK MDA partners. We would like to emphasize that the vision in the paper was written jointly by the UK community as part of the UK NPOP MDA and is not just a collection of strategies of different Institutes. NPOP MDA will also coordinate the implementation of this joint vision. Moreover, at the few instances where the collaboration between UK MDA could be closer, this paper has provided the opportunity to formulate this as part of the community vision. We propose to highlight a lot of this more clearly in the revised version of the paper, e.g. the connection between the theory provided by the University of Reading and its applications by the other partners (including financial mechanisms enabling this), as well as the role of NPOP in jointly implementing the UK MDA community vision.

**3. Lack of novel material compared to existing reviews.**

As already mentioned, we are discussing multiple new and timely topics compared to the existing reviews, such as machine learning and digital twins.  We are also offering an untypically

broad overview of stakeholder applications in MDA and MDA community requirements. This review contains useful detail far beyond the OceanPredict review papers (e.g. on OSSEs and observing missions), which are usually very short. We are happy to better emphasize these novel points in our revised paper and refer to the existing OceanPredict papers to highlight the differences.

**4.        The lack of synthesis in the paper.**

Due to the diversity of MDA areas discussed in this review, it is objectively hard to give a full synthesis across the different areas. This is not necessarily a disadvantage, as addressing the different areas of MDA in the same paper gives the reader an interesting perspective on both commonalities and differences among those areas. The logical synthesis of the vision for each area is presented in the summary section and some synthesis across the different areas in the abstract. We, however, completely agree that more could be done, and we are very happy to revise the summary section to present a deeper and more thorough synthesis, as well as better discussion of the differences in the vision of the different MDA areas.

On top of these main points the reviewer has raised several specific concerns about Figures and some parts of the text, including an interesting point about better linking our proposed developments with the discussed software. We believe these points can be easily addressed in the paper revision and are very happy to do so.

Best wishes,

Jozef Skakala and co-authors

---

## Author Response (AR1)

Dear editor and the reviewers,

Firstly, let us thank both reviewers for their valuable comments. Mainly following the editor recommendation and the second reviewer comments we have done truly major rewriting of the manuscript. The manuscript has been restructured to the extent that it makes little sense to highlight the changes in a separate track-changes file. However, as requested, we are supplying the track-changes file as well, with the sections most impacted by our changes highlighted in blue. The sections not highlighted in blue were also changed, but to a lesser degree. We believe that due to their extent, listing all the changes point by point would make the manuscript completely unreadable, so we would like to thank you for your understanding that such highlights were not provided.

Let us first summarize the most important changes:

1. The manuscript has been completely restructured to provide a more organized, logically coherent and non-repetitive narrative. The sections on theory DA and integrated observing systems have been organically merged into the sections on physics, biogeochemistry and coupled DA (as well as stakeholder section 2.2). The whole science part is now reorganized into section 3 on past and present DA, and section 4 on future vision, both integrating material from all science MDA areas. The section 3 follows the narrative of the newly introduced "workflow diagram" from Fig.2, first outlining how theory is developed at the University of Reading and then describing how it turns into applications within the three MDA areas (physics, biogeochemistry, coupled). These three sections (Section 3.1-3.3) also integrate a lot of the material from the previous sections on observations (see Table 2 and 3) and software (being now closely discussed with the developments), to avoid unnecessary duplication. What remained of the sections on infrastructure (observations, software, hardware and people) became part of the vision section 4. The new structuring of the text enables us to discuss the vision across all MDA areas in a single section 4, contrasting the differences among the needs of different areas (sections 4.1.1-4.1.3), as well as identifying their commonalities (section 4.1.4). This provides a clear logical structure between the different elements of the vision (avoiding the impression of "being a shopping list"). Finally, and very importantly the vision has been now explicitly written from the point of view of the whole UK MDA community, including discussing how it will be jointly pursued, to avoid any impression that it is only a collection of ideas from different partners (criticism by the second reviewer).

2. The new manuscript version highlights much better the close collaboration within the UK community, the roles different Institutes play within this collaboration, as well as explaining how this collaboration developed in the past. To visualize the Institutional roles within the collaborative efforts we have introduced new Figure 2, presenting a "workflow diagram" for how the Institutes developing and running MDA software typically collaborate. The narrative from this Figure is then followed closely by the text in Section 3. Furthermore, the section about the past and present work in the different MDA areas has been rewritten to highlight how the collaboration formed in the past, and how the originally different tools developed at different Institutes mostly converged within a single tool (NEMOVAR). The vision section now includes information on the partners and tools that will turn this vision into reality, further highlighting that this is the collaborative vision of the whole community, rather than specific Institutes. This story is common to all UK MDA areas.

3. Finally, please note that we would like to add an additional author Deep S Banerjee to the manuscript. Although Deep did not do a major contribution to UK MDA during the preparation

of the first draft of the manuscript, during the paper review his contributions as a new member of the UK MDA community became very clear and we now included several of his results into the manuscript. We therefore believe that he deserves co-authorship as well.

In the text below we respond to the comments of the two referees. The referee comment is always in red, our response in blue and the cited text in green and italic.

**Reviewer 1:**

We thank the reviewer for their comments and their positive evaluation. We have already responded to the reviewer point by point in our public response at Egusphere. Here we will just summarize the response and update the line numbers, as the text has been reshaped again due to major revisions that happened after the second review was received.

1. L 110; L115; L135; L150, L630-660; L670 : references could be added to ensure consistency with the rest of the article.

We thank the reviewer, the following references/links have been added to address the issues:

**L110:** here we don't have a specific reference, but the following weblink is provided: *"https://www.cefas.co.uk/science/emergency-response/"*

**L115:** the following reference is provided:

Guiavarc'h, C., Roberts-Jones, J., Harris, C., Lea, D.J., Ryan, A. & Ascione, I. (2019) Assessment of ocean analysis and forecast from an atmosphere-ocean coupled data assimilation operational system. Ocean Science, 15, 1307–1326. Available from: https://doi.org/10.5194/os-15-1307-2019

**L135:** the following references are provided:

[revised manuscript text omitted]

2. L340, L880 : EO has not been introduced, and it seems a little odd to mention the Earth Observation community only there in the summary. It could also be explicitly mentioned in the paragraphe on observing systems design L893

We thank the reviewer for spotting this and we have introduced EO already on line 364. The text has been completely rearranged, so there is no longer the paragraph with the line number 893, but we have explicitly mentioned EO on line 498, in the similar context as the original paragraph around line 893 (mentioned by the reviewer).

3. What we see on Fig 2 is not really explained. What is error 242? (see title of the figure), does blue mean an error reduction?

The "242" in the Figure title was a typo and has been removed. We have changed the Figure caption to better explain the content of the Figure (now Fig.3) as follows:

*"Change in temperature Root Mean Square Errors (RMSE) between two experiments. The reference experiment uses parameterized background covariance matrix model, the second experiment uses the ensemble-based hybrid background covariance matrix model. Temperature RMSE is computed using model short-range forecasts against all in-situ observations in the upper*

*200 m (left) and in the 200-1000 m range (right), for year 2017. Negative values show improvement when using ensemble-based hybrid background covariance model."*

4. L698: Kd has not been introduced

   Thank you, we have now introduced it, please see Table 3.

5. L244 : It may be easier to follow by adding one sentence, maybe on the time scales of the impact of the atmospheric forcing.
   Thank you, we have addressed this by the text below (please see lines 458-461):

*"Both waves and surge models are highly influenced by the wind (waves, surge) and atmospheric pressure (surge). Saulter et al. (2020) showed that assimilation of data into a regional wave model using NEMOVAR improved the forecasts over lead times of up to 12 hours, but errors in the surface forcing and wave model parameterisations dominated the forecast errors beyond 1-2 day lead time. The Met Office thus plans to develop capability to...."*

6. L426-427 : a couple of sentences could explain what we see on Fig4.

   Thank you, we have added the following text to the former Fig.4, now Fig.5, caption (we think it is very specific information to be discussed in the main part of the text):

*"In the tropics we see negative correlations associated with warm SST and low wind speeds, which are linked to diurnal variations in solar radiation, with correlations strengthening as the ocean surface warms throughout the day. In contrast, the significant positive correlations of SST with 10 m wind speed in the North Atlantic are linked with strong SST gradients and tend to be associated with areas of stronger winds, the location of which varies synoptically. These areas of larger ocean/atmosphere correlations in mid-latitudes were shown to extend vertically into the ocean, throughout the mixed layer (Wright et al., 2024)."*

7. L455: "the combined impact of physics and BGC DA is also dependent on the assimilation methodology": maybe one or two examples would help to link with future tests and potential improvements.

   We expanded the sentence as follows (see the lines 404-407):

*"The combined impact of physics and BGC DA is also likely to be dependent on the assimilation methodology (Nerger et al, 2023), for instance how alignment of fronts and other features is considered (Anderson et al., 2000; Yu et al., 2018), and how increments of different variables are projected onto different scales (Waters et al., 2017)."*

8. L 473: as there are slightly more details about this in 3.2.2 the authors may refer to this section

   Thank you, we have done this.

9. L1040 : inconsistency between Eyre et al (2021) in the text, and (2022) in the references

10. L 1188 : add the year in reference Nerger et al (2023)

Thank you, this has been now addressed.

**Reviewer 2:**

We thank the reviewer for their comments. Please find the point by point response below:

1.  The past and present are quite well covered by the manuscript but the most interesting -the future - is yet mysterious in the manuscript.

As already mentioned in the opening part of our response letter (page 1 of the letter), we have now rewritten the vision part into a single section and substantially improved upon the clarity and the logical structure of the vision. The new structure helps to (i) emphasize that this is a common vision of the UK MDA community, and is highly collaborative (in fact in the whole scientific vision section 4.1, there is only one example of a case where only one partner is involved, i.e. on the lines 448-449), (ii) make sure we can highlight both the differences and similarities among the different MDA areas. By doing so we include a range of important topics, such as high-resolution DA, multivariate DA, parameter estimation, strongly coupled DA, ML-DA, smoothers and digital twins.

2.  Even though there is no denying that the UK has contributed - and is still contributing - pioneering efforts in marine data assimilation, I cannot see the purpose of publishing a national perspective on marine data assimilation strategy in an international scientific journal. A possible benefit for readers would have been if the UK proposed a way forward that was applicable to all other nations, as if the UK were a laboratory where an efficient collaboration between universities, the national marine institutes and the weather forecasts centres followed a common strategy and a detailed implementation plan. However I failed to see signs of coordination in the manuscript that are beyond the status quo today.

and

Six UK institutes are actively using marine data assimilation in the UK, according to Table 1. Two institutes are operational weather forecasting centres (UK Metoffice and ECMWF) and four institutes use DA for research. This is an impressive number for a single nation and the question naturally arises of how these institutes will cooperate and build up synergies despite their respective institutional constraints.

Firstly, it is hopefully now much clearer that UK MDA is a very closely collaborating entity, this has been in the new manuscript version highlighted in the many ways described at the start of this letter (page 1, mainly point 2), including the new Fig.2. At present, we have already a very

close collaboration within the whole UK MDA (as highlighted in Fig.2), and this collaboration should be certainly maintained and even expanded in the future, as is now clearly emphasized in the vision section (section 4). Furthermore, we believe the amount of detail provided in the vision (section 4) is optimal, any more detailed implementation plan would make the paper unnecessarily long and tiring to read (the science part of vision section is already 4 and half pages).

As we mentioned in the initial response to reviewer comments published at Egusphere, for a closely collaborating entity, such as UK MDA, it makes very good sense to write a review and vision paper and publish it in an international journal (for examples see Institute-specific papers published in the same journal, such as Siddorn et al, 2016, https://os.copernicus.org/-articles/12/217/2016/). Similarly, as already discussed in our initial response, this review enables us to go into much greater detail than the OceanPredict reviews (which are typically short). Unlike those reviews we compare the full range of MDA areas (physics, biogeochemistry, coupled), their different needs and commonalities, address new topics such as ML and digital twins and dive much more deeply into stakeholder applications, with the aim to provoke discussion with other scientific and non-scientific communities.

The reasons for this review are now also briefly discussed in the Introduction (paragraph around line 55), mentioning also the OceanPredict reviews by Moore et al, 2019, Fennel et al, 2019, Martin et al, 2025:

*"UK MDA is a closely collaborating community with the collaboration largely facilitated by the UK National Partnership for Ocean Prediction (NPOP) and its MDA group. The role of this paper prepared by the NPOP MDA group is to both highlight the history of the ever-increasing collaboration within the UK community and formulate a unified vision for future developments. Whilst the paper is UK-focused, it should also be of interest to the broader international community, since it provides both a useful example of a successful national collaboration, and the UK vision will feed into international MDA developments due to the UK's leading role in this area. Furthermore, such focus allows topics to be explored with greater detail and synergy, complementing relatively recent international community reviews (e.g. Moore et al., 2019; Fennel et al., 2019, Martin et al, 2025), as well as going beyond by discussing topics that emerged since some of those reviews were written. "*

3. A cornerstone of collaboration efforts is the use of common tools. This is not compulsory for scientists who can exchange ideas, but leads to significant economies of scale when the software needs a multidisciplinary team for maintenance, for example the European NEMO ocean modeling engine that has been successfully adopted by most UK institutes. So I would have expected the paper to announce a software development plan that would accommodate all the wishes from all institutes. There is indeed an intention to reduce the number of softwares from four to two, one optimised for operations and one simpler for research, but that information comes very late in the paper (Section 4.2) so it is too late to sort out which ideas will fit in which softwares and how. In other words, there is no research-to-operations plan in the manuscript, and the reader is left to imagine the sequence of steps that will incorporate the numerous ideas listed in the manuscript into a common tool.

This has been now fully addressed in sections 3 and 4. We apologize for not making this point obvious in the previous version of the manuscript: indeed almost all proposed developments are within the broader framework of NEMOVAR (with the only exception being parameter estimation in marine biogeochemistry DA). Note that moving towards unification of software is also planned in the near future for air-sea coupled DA, where NEMOVAR will be used in synergy with JEDI/OOPS (for discussion, see section 4.1.3). We have made this very clear in the new version of the manuscript. Also we now consistently link each development mentioned within the section 3 to the corresponding software, so the tools are discussed as integral part of the science.

4. About the main contents of the paper, the authors have chosen to provide either long lists of bullet points or short sections exposing individual ideas to be followed up. These bullet points and ideas read as shopping lists without logical links: each item apparently provided by one institution or another. One example is section 3.3.2 (coupled DA) that ends on an enumeration of four steps that mix scientific and technical considerations, ending with (iv) that does not seem connected with the previous three. There are also repetitions between sections (for example lines 204 to 209 are repeated from section 2.2). As such, the manuscript makes a tedious read and does not provide a positive example of consolidated cross-institutional collaboration.

This has been now completely rewritten. As already stated at the start of the letter, and elsewhere, basically all the goals listed in the vision section 4 (see especially 4.1.1-4.1.4) are collaborative, and based on the same type of software (NEMOVAR). We believe the logical links are now firmly established throughout the text. As for the coupled DA section (former section 3.3.2, now 4.1.3), this has been hugely simplified and is written very differently, making a general case for strongly coupled DA that spans across the different types of coupling (for details see section 4.1.3). However, we still use bullet points wherever it makes the text more transparent and benefits the reader, such as in the stakeholder sections 2.1-2.2 and at the start of section 3.

5. Loose ends are yet another weakness of the manuscript. One example among many others can be found on line 360: "More complex balance relationships could use as their starting point the mass conservation scheme of Hemmings et al (2008)". Why this one in particular? "or focus on ML/statistical modelling, which is being pursued in a current studentship between PML and the University of Reading". This is blue sky to the reader: It does not say why ML/statistical modelling is relevant here, how much can be expected from a studentship or if there are other options on the table than these two.

During the rewriting of the manuscript we have fixed such cases, however not every example can be given a comprehensive explanation due to lack of space (references are always available though). The particular examples given by the reviewer have been updated as (see pages 22-23, lines 475-479):

*"An essential goal for BGC MDA is to develop more reliable multi-variate techniques. It is planned to assess both balancing and ensemble approaches, at first separately and then in combination. These can be based on the ensemble-NEMOVAR (3DEnVar) developments from Lea et al 2022 and Skakala et al 2024, through adapting and/or expanding existing balancing schemes (e.g. Hemmings et al, 2008), or using ML to learn relationships between observed and unobserved variables (as explored in Higgs et al, 2025, in prep)."*

6. The wishlists relating to data assimilation methodology can be found in international community papers as the issues raised are not more specific to UK marine research than anywhere else. Therefore, contributions to international community papers such as those issued by the OceanPredict community every four years have a larger impact on the community and these almost always receive inputs from the UK institutes involve here.

The reasons why we think that this review is an important addition to the existing literature have been already stated a number of times. In this particular aspect we believe that our vision, spanning across all MDA topics, is much more detailed than what is available in the existing OceanPredict reviews.

7. The explanations are often missing. Here is another example: "Simple post-processing balances will be applied in the next ECMWF systems such that sea ice increments will induce near surface ocean temperature increments, but not the other way around." Why is that? "Similar plans are being considered at the Met Office." This is another very vague indication, is the reader expected to take action of that?

Similarly to comment 5, these have been now fixed during the rewriting of the manuscript. In the particular case mentioned by the reviewer, the text has been removed, as we have considered the level of detail given by the paragraph unnecessary.

8. The Figures are taken out of their context from already published material and are exposed without explanations. For example Fig. 4 is a map of correlations between ocean and atmospheric variables, intended to convey the point that coupled assimilation is a good thing. However, without a minimum of explanations, it is not possible to tell whether these values of correlation are realistic or not. The same point can be made about all the other figures. Figure 1 - a map of the institutes in the UK - does not provide any additional information above the list of authors affiliations.

The main reasons behind the Figures are twofold: (i) to strengthen and visualise certain important points expressed in the text (this is the case of Fig.1-2) and (ii) to give examples of discussed research, to increase the appeal of the paper to the reader (this is the case of Fig.3-5). It is hard to imagine any significantly different roles the Figures could play in this type of review paper. We believe that although Figures of the first type do not necessarily reveal new information, they are still highly valuable to visually communicate the key points of the paper (in case of Fig.1 it is to show the key contributors to UK MDA, not just the author affiliations). The Figures of the second type can always be thought to be slightly out of context, but we believe in their value too (as per our point (ii) at the start of this answer). However, we agree that the content of each Figure needs to be adequately explained in the caption (even though the finest details might be missing due to lack of space and can be always found in the cited references). We apologize that this wasn't the case in the previous version of the manuscript and have made now sure that the information in each Figure caption is sufficiently complete. We also understand that the appropriateness of using Figures in this type of manuscript might depend on an individual opinion, and some shorter papers (e.g. some of the OceanPredict reviews) seem not to use Figures at all. However, unless there is a strict requirement to abandon the Figures, we would like to maintain them as they are, if possible (especially Fig.1-2).

9. The only interestingly novel statements can be found about the implementation of advanced Fortran softwares NEMO and NEMOVAR on GPUs, which would be very useful ways to reduce the energy consumption of marine data assimilation systems. Unfortunately these strong claims are not substantiated, would it only be by some indications of computational efficiency, so I would take them with a pinch of salt.

As already mentioned, we sincerely believe that there is much more novelty in the paper than implementation of Fortran code on GPUs. With this comment we assume the reviewer refers to what is now Section 4.2.3, lines 664-672. We have to however admit, we do not fully understand what type of evidence the reviewer requests in this comment and which "strong claims" the reviewer exactly means. In any case, as stated in the text on the lines 664-671, substantial amount of work has been already completed towards porting NEMO and NEMOVAR on the GPUs, involving the PSyclone software, so the task is quite achievable and exciting. We are unsure it would be desirable to provide more detail on this topic, given the broad scope of the paper, but if requested we could do so.

10. The Digital Twin of the Ocean (DTO), an emerging concept in the domain, is first cited out of the blue in line 490, without context. This is another missed occasion to provide a singular reflexion from the perspective of prominent UK institutes.

Digital twin is indeed an important growing topic significantly overlapping with MDA. We have briefly mentioned them in the Introduction, scientific stakeholder section 2.2, and biogeochemistry MDA (section 3.2). We dive deeper into digital twins in the vision section (section 4), because although nice examples of digital twins in UK MDA already exist, this is still largely an emerging concept for the future. We dedicate to digital twins a long paragraph in section 4.1.4 (lines 559-583), providing a description of the concept, the UK collaboration working on this topic (including Met Office, NOC, PML, University of Exeter), noting the UK work already done in this area, and ideas of how this work should be developed in the future (e.g. discussing and explaining the role of ML/AI). This section provides the context and the reflection that the reviewer wanted to see, within the space allowed by the review paper, addressing otherwise much broader topics. Finally, please note, that it is quite possible there is misunderstanding between us and the reviewer coming from the fact that the word "digital twin" is sometimes used more broadly than how we understand it in the paper, but our understanding is consistent with the definitions used by the UN Decade DiTTO and European DTO.

We would like to thank again both the reviewers for their comments that significantly improved the manuscript. We hope that after the major revisions the paper will be suitable for publication in Ocean Science.

Best wishes,

Jozef Skakala and co-authors

---

## Author Response (AR2)

We would like to sincerely thank both the reviewers for their positive comments about the paper and for their valuable suggestions on how to further improve it. These suggestions are addressed below.

The reviewer comment is always in blue, our response in black and the cited change in green and italic. Please note that the changed sections are also marked in blue font in the submitted track-changes version of the manuscript.

**Reviewer 1:**

The review mentioned and discussed collaboration in the UK and its national perspective. Then the review, I think, has useful information to oceanographic community in other countries. I recommend the publication of the review after the following very minor revision.

Minor Comments:

2.1 End-user applications: I think recent environmental DNA (eDNA) observation is very important for understanding of biodiversity. If we try to analyze space-time distribution of eDNA with the MDA information, it would be very progressive. Then please try to discuss/add a paragraph about the MDA application to eDNA analyses such as:
• Marine ecosystem and biodiversity. The use of environmental DNA (eDNA) is recently used for studying the ecosystem and variability of life in the sea as a biodiversity observation network. The space-time distribution of eDNA is relatively new but growing rapidly. The distribution would be greatly useful with ocean physical and biogeochemical information by MDA.

We thank the reviewer for this suggestion. We have indeed added the biodiversity (and eDNA) application to section 2.2 (rather than 2.1 as we believe this is a scientific application). Please see the last point on the page 9, lines 210-215:

*"- reanalysis data can be used for interpreting drivers of change seen in biodiversity datasets, such as from the Continuous Plankton Recorder, e.g. see the work of Holland et al., (2024). Looking more into the future, reanalyses have also the potential to assist with interpreting newer, rapidly growing datasets including those based on environmental DNA (eDNA)."*

LL566-589 paragraph: An example of the use of AUV is discussed. I agree with the use of high-spatial model resolution. I think the model would/should be "a relocatable model?" And I would have expected the paper to mention an "on-spot DA" around AUV very local position? Or very many AUVs observation data would be assimilated?

Again, we thank the reviewer for this suggestion. The AUV assimilation considers data from both a single, or multiple, AUVs, depending on the observational set-up. The data are usually suitably

spatially and temporally thinned, with the model data mapped into the corresponding AUV positions via observational operators. The discrepancy in spatial and temporal scale between the AUVs and the model is partly addressed through the thinning, but to a degree remains a challenge to be addressed in the future. These points are discussed in the referenced literature (e.g. Skakala et al, 2021, Ford et al, 2022). The model can be covering a larger area (e.g. the whole NWES as in Partridge et al, 2025) in its application, but could be in the future also made relocatable for higher resolution. We have added some sentences on this, please see the page 27, line 605-610:

*"The biogeochemistry high-resolution models could cover large areas, such as the whole NWES (as Partridge et al, 2025, in prep), or could cover only smaller areas in the coastal zone. In such case they could be made relocatable, e.g. following techniques developed by Shapiro et al (2022, 2023)."*

**Reviewer 2:**

I found this review to be insightful and highly informative. It offers a clear overview of the current landscape of marine data assimilation (MDA) and outlines a compelling vision for future developments, particularly in the areas of coupled systems, digital twins, and the application of machine learning.

One section that may benefit from clarification is the discussion of weakly coupled air-sea systems. In particular, the phrase "ocean ensemble developments (p.19)" is somewhat vague; it is not immediately clear whether this refers to ensemble-based data assimilation, ensemble forecasting, or both. In addition, it would help to clarify what variables these developments aim to control—presumably the initial state of the ocean (e.g., temperature, salinity), but this is not made explicit.

We thank the reviewer for this comment, we have now hopefully made these points clearer through the updated text in the section 3.3, please see the page 20, line 405:

*"Weakly coupled air-sea systems are part of ECMWF (de Rosnay et al, 2022) and Met Office (Lea et al, 2015, Guiavarc'h et al., 2019) operational short-range weather forecasts. The ocean part of the Met Office coupled NWP ensemble system is currently being developed to include improved ensemble forecast generation methods and the use of hybrid-3DEnVar (Lea et al., 2022; 2023). This will allow improved uncertainty propagation from the ocean to the atmosphere through the forecast, leading to improved forecast uncertainties in both ocean and atmosphere, and should also enable improvements in the accuracy of the ocean physical variables."*

The definition of the digital twin is included (p.26), but it is somewhat embedded within the narrative. I suggest presenting it more prominently—perhaps as a standalone sentence early in the relevant section—and briefly connecting it to the practical examples (e.g., AUV navigation). For non-expert readers, a short plain-language summary of the loop (e.g., "the model tells the AUV where to go next, and the AUV's new observations help update the model") could also be helpful. A simple schematic illustrating this feedback loop would greatly enhance clarity.

We thank the reviewer for this great suggestion, we have now made this much more prominent by including a new Figure 5 (see the page 17) showing schematic representation of the digital twin system used in Ford et al (2022), and also in Partridge et al (2025). The Figure caption provides a simple explanation of how the digital twin works, along the lines suggested by the reviewer:

*"**Fig.5.** A schematic illustration of a digital twin system navigating fully autonomous gliders to areas of observational interest. The Figure is reproduced from Ford et al (2022), with a similar scheme also being applied in Partridge et al (2025), in prep. The digital twin system is based on information flowing in all directions: (i) glider observations are being assimilated into the pre-operational forecasting model (i.e. the model is updated by the glider). (ii) The operational model subsequently produces forecasts for a stochastic/ML model, with additional inputs into the stochastic model provided by the glider directly, (iii) The stochastic model then provides the system with a fully autonomous path-planning capacity close to the glider's spatial scale of operations, navigating the glider into the expected areas of observational interest (i.e. the model tells glider "where to go"). This exchange of information then cycles throughout the glider mission. "*

More broadly, the discussion of strongly coupled data assimilation (SCDA) might benefit from briefly touching on the role of time-ordered information flow across components. While the paper rightly highlights the challenge of different temporal scales between ocean and atmosphere, an even richer perspective might come from recent mathematical tools—such as path signatures (Lyons, 1998) and signature kernels (Chevyrev & Oberhauser, 2016)—which are designed to extract time-ordered moments from multivariate path data. These representations preserve the temporal structure and non-commutative properties of the path, and may offer future opportunities to improve the modeling of cross-component covariances and adjoint sensitivities in SCDA.

We thank the reviewer for mentioning these approaches, we have now added a sentence (along the lines suggested by the reviewer) to the section 4.1.3 on the coupled DA vision mentioning them (and the relevant references). Please see the line 545, page 25:

*"The community should in future be open to new methods to advance strongly coupled DA, such as those based on path signatures (Lyons, 2014) and signature kernels (Chevyrev & Oberhauser, 2018), which are designed to extract time-ordered moments from multivariate path data."*

Finally, I appreciate that this manuscript uses "digital twin" in a relatively narrow and rigorous sense—that is, referring to real-time coupled systems with feedback into observation strategy. Given the broader and often ambiguous usage of this term across domains, it might be helpful to briefly acknowledge this and clarify that the review is focused on a specific, operationally meaningful subset of digital twin applications. This would help avoid conceptual drift and assist readers in understanding the scope of the discussion.

We acknowledge that our use of the digital twin concept is rather more specific than in some of the literature. To clarify this for the reader we will propose a footnote in the Introduction (line 85, page 3), with the following:

*"Digital twins are understood here in a quite specific sense, as systems where the digital twin interacts two-way (exchanging information both-ways) with the twinned physical object, whilst operating as a real time decision-making tool. The fully autonomous observing systems described here fullfil this operational definition of digital twin."*

These suggestions are offered with appreciation for an already excellent review. I believe that with minor clarifications, the manuscript will serve as a very useful and timely resource for the MDA community and beyond.

We would like to thank both reviewers for their valuable input. We hope that the paper is now ready to be accepted for publication in OS.

Best wishes,
Jozef Skakala and the co-authors